# Neural network analysis of sleep stages enables efficient diagnosis of narcolepsy

Jens B. Stephansen[1,2], Alexander N. Olesen[1,2,3], Mads Olsen[1,2,3], Aditya Ambati [1], Eileen B. Leary [1], Hyatt E. Moore[1], Oscar Carrillo[1], Ling Lin[1], Fang Han[4], Han Yan[4], Yun L. Sun[4], Yves Dauvilliers[5,6], Sabine Scholz[5,6], Lucie Barateau[5,6], Birgit Hogl[7], Ambra Stefani[7], Seung Chul Hong[8], Tae Won Kim[8], Fabio Pizza[9,10], Giuseppe Plazzi[9,10], Stefano Vandi[9,10], Elena Antelmi[9,10], Dimitri Perrin[11], Samuel T. Kuna[12], Paula K. Schweitzer[13], Clete Kushida[1], Paul E. Peppard[14], Helge B.D. Sorensen[2], Poul Jennum[3] & Emmanuel Mignot[1]

Analysis of sleep for the diagnosis of sleep disorders such as Type-1 Narcolepsy (T1N) currently requires visual inspection of polysomnography records by trained scoring technicians. Here, we used neural networks in approximately 3,000 normal and abnormal sleep recordings to automate sleep stage scoring, producing a hypnodensity graph—a probability distribution conveying more information than classical hypnograms. Accuracy of sleep stage scoring was validated in 70 subjects assessed by six scorers. The best model performed better than any individual scorer (87% versus consensus). It also reliably scores sleep down to 5 s instead of 30 s scoring epochs. A T1N marker based on unusual sleep stage overlaps achieved a specificity of 96% and a sensitivity of 91%, validated in independent datasets. Addition of HLA-DQB1*06:02 typing increased specificity to 99%. Our method can reduce time spent in sleep clinics and automates T1N diagnosis. It also opens the possibility of diagnosing T1N using home sleep studies.

[1] Center for Sleep Science and Medicine, Stanford University, Stanford 94304 CA, USA. [2] Department of Electrical Engineering, Technical University of Denmark, Kongens Lyngby 2800, Denmark. [3] Danish Center for Sleep Medicine, Rigshospitalet, Glostrup 2600, Denmark. [4] Department of Pulmonary Medicine, Peking University People's Hospital, Beijing 100044, China. [5] Sleep-Wake Disorders Center, Department of Neurology, Gui-de-Chauliac Hospital, CHU Montpellier 34295, France. [6] INSERM, U1061, Université Montpellier 1, Montpellier 34090, France. [7] Department of Neurology, Innsbruck Medical University, Innsbruck 6020, Austria. [8] Department of Psychiatry, St. Vincent's Hospital, The Catholic University of Korea, Seoul 16247, Korea. [9] Department of Biomedical and Neuromotor Sciences, University of Bologna, Bologna 40123, Italy. [10] IRCCS Istituto delle Scienze Neurologiche di Bologna, Bologna 40139, Italy. [11] School of Electrical Engineering and Computer Science, Queensland University of Technology, Brisbane 4001, Australia. [12] Department of Medicine and Center for Sleep and Circadian Neurobiology, University of Pennsylvania, Philadelphia 19104 PA, USA. [13] Sleep Medicine and Research Center, St. Luke's Hospital, Chesterfield 63017 MO, USA. [14] Department of Population Health Sciences, University of Wisconsin-Madison, Madison 53726 WI, USA. These authors contributed equally: Jens B. Stephansen, Alexander N. Olesen. These authors jointly supervised this work: Helge B.D. Sorensen, Poul Jennum, Emmanuel Mignot. Correspondence and requests for materials should be addressed to E.M. (email: mignot@stanford.edu)

Sleep disorders and sleep dysregulation impact over 100 million Americans, contributing to medical consequences such as cardiovascular (arrhythmia, hypertension, stroke), metabolic (diabetes, obesity) and psychiatric disorders (depression, irritability, addictive behaviors). Sleep deprivation impairs performance, judgment and mood, and is a major preventable contributor to accidents[1]. There are ~90 different sleep disorders including insomnia (20% of population), obstructive and central sleep apnea (10%), restless legs syndrome (4%), rapid eye movement (REM) sleep behavior disorder (RBD) and hypersomnia syndromes such as type 1 narcolepsy (T1N)[2].

Among these pathologies, T1N is unique as a disorder with a known, discrete pathophysiology—a destruction of hypocretin neurons in the hypothalamus likely of autoimmune origin[3]. This is reflected in the cerebrospinal fluid (CSF) concentrations of the hypocretin-1 (orexin-A) neuropeptide, where a concentration below 110 pg/ml is considered indicative of narcolepsy[2]. Typically beginning in childhood or adolescence, narcolepsy affects approximately 0.03% of the US, European, Korean and Chinese populations[4]. Unique to narcolepsy is the extremely strong (97% versus 25%) association with a genetic marker, HLA-DQB1*06:02[5], and a well-characterized set of sleep disturbances that include short sleep latency, rapid transitions into REM sleep and poor nocturnal sleep consolidation. The pathology also includes episodes of "sleep/wake dissociation" where the patient is half awake and half in REM sleep, for example, experiencing REM sleep muscle paralysis while awake (sleep paralysis, cataplexy) or dreaming while awake (hypnagogic hallucinations).

Sleep disorders are generally assessed at sleep clinics by performing sleep analysis using nocturnal polysomnography (PSG), a recording comprised of multiple digital signals which include electroencephalography (EEG), electrooculography (EOG), chin and leg electromyography (EMG), electrocardiography, breathing effort, oxygen saturation and airflow[6]. When sleep is analyzed in PSGs, it is divided into discrete stages: wake, non-REM (NREM) sleep stage 1 (N1), 2 (N2) and 3 (N3), and REM. Each stage is characterized by different criteria, as defined by consensus rules published in the American Academy of Sleep Medicine (AASM) Scoring Manual[6,7]. N1 (sleep onset) is characterized by slowing of the EEG, disappearance of occipital alpha waves, decreased EMG and slow rolling eye movements, while N2 is associated with spindles and K-complexes. N3 is characterized by a dominance of slow, high amplitude waves (>20%), while REM sleep is associated with low voltage, desynchronized EEG with occasional saw tooth waves, low muscle tone and REMs. PSG analysis is typically done by certified technicians who, through visual inspection on a standardized screen, assign a sleep stage to each 30 s segment of the full recording. Although there is progression from N1 to N3 then to REM during the night, a process that repeats approximately every 90 min (the sleep cycle), each stage is associated with physiological changes that can be meaningful to the assessment of sleep disorders such as obstructive sleep apnea. For example, the abnormal breathing events that occur with obstructive sleep apnea (OSA) are generally less severe in N3 versus N2 because of central control of breathing changes, and they are more severe in REM sleep, due to upper airway muscle weakness[8]. The differentiation of sleep stages is also particularly important for the diagnosis of narcolepsy, a condition currently assessed by a PSG followed by a multiple sleep latency test (MSLT), a test where patients are asked to nap 4–5 times for 20 min every 2 h during the daytime and sleep latency and the presence of REM sleep is noted[9]. A mean sleep latency (MSL) less than 8 min (indicative of sleepiness) and the presence of at least 2 sleep onset REM periods (SOREMPs, REM latency ≤15 min following sleep onset in naps) during the MSLT or 1 SOREM plus a REM latency ≤15 min during nocturnal PSG is diagnostic for narcolepsy. In a recent large study of the MSLT[10], specificity and sensitivity for type 1 narcoleptics were, respectively, 98.6% and 92.9% in comparing 516 T1N versus 516 controls and 71.2% and 93.4% in comparing 122 T1N cases versus 132 other hypersomnia cases (high pretest probability cohort). Similar sensitivity (75–90%) and specificity (90–98%) have been reported by others in large samples of hypersomnia cases versus T1N[11–15].

Manual inspection of sleep recordings has many problems. It is time consuming, expensive, inconsistent, subjective and must generally be done offline. In one study, Rosenberg and Van Hout[16] found inter-scorer reliability for sleep stage scoring to be 82.6% on average, a result consistently found by others[17–20]. N1 and N3 in particular have agreements as low as 63 and 67%, placing constraints on their usefulness[16]. In this study, we explored whether deep learning, a specific subtype of machine learning, could produce a fast, inexpensive, objective, and reproducible alternative to manual sleep stage scoring. In recent years, similar complex problems such as labeling images, understanding speech and translating language have seen advancement to the point of outperforming humans[21–23]. Several high-profile papers have also documented the efficacy of deep learning algorithms in the healthcare sector, especially in the fields of diabetic retinopathy[24,25], digital pathology[26,27] and radiology[28,29]. This technology refers to complex neural network models with a very large number (on a magnitude of millions) of parameters and processing layers. For a thorough review of the underlying theory behind deep learning including common model paradigms, we refer to the review article by LeCun et al.[30].

In this implementation of deep learning, we introduce the hypnodensity graph—a hypnogram that does not enforce a single sleep stage label, but rather a membership function to each of the sleep stages, allowing more information about sleep trends to be conveyed, something that is only possible in non-human scoring. Using this concept, we next applied deep learning-derived hypnodensity features to the diagnosis of T1N, showing that an analysis of a single PSG night can perform as well as the PSG-MSLT gold standard, a 24 h long procedure.

## Results

**Inter-scorer reliability cohort.** Supplementary Table 1 reports on the description of the various cohorts included in this study, and how they were utilized (see Datasets section in Methods). These originate from seven different countries. We assessed inter-scorer reliability using the Inter-scorer Reliability Cohort (IS-RC)[31], a cohort of 70 PSGs scored by 6 scorers across three locations in the United States[31]. Table 1 displays individual scorer performance as well as the averaged performance across scorers, with top and bottom of table showing accuracies and Cohen's kappas, respectively. The results are shown for each individual scorer when compared to the consensus of all scorers (biased) and compared to the consensus of the remaining scorers (unbiased). In the event of no majority vote for an epoch, the epoch was counted equally in all classes in which there was disagreement. Also shown in Table 1 is the model performance on the same consensus scorings as each individual scorer along with the t-statistic and associated p value for each paired t-test between the model performance and individual scorer performance. At a significance level of 5%, the model performs statistically better than any individual scorer both in terms of accuracy and Cohen's kappa.

Supplementary Table 2 displays the confusion matrix for every epoch of every scorer of the inter-scorer reliability data, both unadjusted (top) and adjusted (bottom). As in Rosenberg and Van Hout[16], the biggest discrepancies occur between N1 and

**Table 1 Individual and overall scorer performance, expressed as accuracy and Cohen's kappa**

|  | Overall | Scorer 1 | Scorer 2 | Scorer 3 | Scorer 4 | Scorer 5 | Scorer 6 |
|---|---|---|---|---|---|---|---|
| Accuracy (%), biased | 81.3 ± 3.0 | 82.4 ± 6.1 | 84.6 ± 5.5 | 74.1 ± 7.9 | 85.4 ± 5.7 | 83.1 ± 9.4 | 78.3 ± 8.9 |
| Accuracy (%), unbiased | 76.0 ± 3.2 | 77.3 ± 6.3 | 79.1 ± 6.3 | 69.0 ± 8.0 | 79.7 ± 6.5 | 77.8 ± 9.6 | 72.9 ± 9.2 |
| Model accuracy (%) on concensus | — | 85.1 ± 4.9 | 83.8 ± 5.0 | 86.5 ± 4.3 | 84.3 ± 4.7 | 85.6 ± 4.7 | 87.0 ± 4.5 |
| T-stat (p value) | — | 9.5 ($3.8 \times 10^{-14}$) | 6.6 ($7.5 \times 10^{-9}$) | 18.3 ($6.0 \times 10^{-28}$) | 6.7 ($4.7 \times 10^{-9}$) | 6.4 ($1.7 \times 10^{-8}$) | 12.2 ($7.5 \times 10^{-19}$) |
| Cohen's kappa, biased | 61.0 ± 6.8 | 63.6 ± 12.2 | 68.4 ± 10.5 | 45.6 ± 19.7 | 69.6 ± 13.2 | 64.5 ± 20.9 | 54.5 ± 19.8 |
| Cohen's kappa, unbiased | 57.7 ± 6.1 | 61.3 ± 11.2 | 64.6 ± 10.3 | 43.5 ± 19.2 | 64.6 ± 13.1 | 60.9 ± 16.9 | 51.6 ± 16.7 |
| Model kappa on concensus | — | 74.3 ± 12.3 | 72.4 ± 12.1 | 76.0 ± 11.8 | 72.7 ± 12.0 | 74.7 ± 12.1 | 76.6 ± 12.2 |
| T-stat (p value) | — | 9.5 ($4.6 \times 10^{-14}$) | 7.1 ($7.9 \times 10^{-10}$) | 15.4 ($7.0 \times 10^{-24}$) | 6.6 ($6.4 \times 10^{-9}$) | 7.1 ($9.2 \times 10^{-10}$) | 13.2 ($2.0 \times 10^{-20}$) |

Both accuracy and Cohen's kappa are presented as both with (biased) and without (unbiased) the assessed scorer included in the consensus standard in a leave-one-out fashion. Accuracy is expressed in percent, and Cohen's kappa is a ratio, and therefore unitless. T-statistics and p values correspond to the paired t-test between the unbiased predictions for each scorer against the model predictions on the same consensus

Wake, N1 and N2, and N2 and N3, with some errors also occurring between N1 and REM, and N2 and REM.

For future analyses of the IS-RC in combination with other cohorts that have been scored only by one scorer, a final hypnogram consensus was built for this cohort based on the majority vote weighted by the degree of consensus from each voter, expressed as its Cohen's $\kappa$, $\kappa = 1 - \frac{1-p_o}{1-p_e}$, where $p_e$ is the baseline accuracy and $p_o$ is the scorer accuracy, such that

$$\mathbf{y} = \arg \max \frac{\sum_{i=1}^{6} \widehat{\mathbf{y}}_i \cdot \boldsymbol{\kappa}_i}{\sum_{i=6}^{6} \boldsymbol{\kappa}_i}. \quad (1)$$

In this implementation, scorers with a higher consensus with the group are considered more reliable and have their assessments weighted heavier than the rest. This also avoided split decisions on end-results.

**Optimizing machine learning performance for sleep staging**. We next explored how various machine learning algorithms (see Methods) performed depending on cohort, memory (i.e., feed forward (FF) versus long short-term memory networks (LSTM)), signal segment length (short segments of 5 s (SS) versus long segments of 15 s (LS)), complexity (i.e., low (SH) vs. high (LH)), encoding (i.e., octave versus cross-correlation (CC) encoding, and realization type (repeated training sessions). The performance of these machine learning algorithms was compared with the six-scorer consensus in the IS-RC and with single scorer data in 3 other cohorts, the Stanford Sleep Cohort (SSC)[10,32], the Wisconsin Sleep Cohort (WSC)[32,33] and the Korean Hypersomnia Cohort (KHC)[10,34] (see Datasets section in Methods for description of each cohort).

Model accuracy varies across datasets, reflecting the fact scorer performance may be different across sites, and because unusual subjects such as those with specific pathologies can be more difficult to score—a problem affecting both human and machine scoring. In this study, the worst performance was seen in the KHC and SSC with narcolepsy, and the best performance was achieved on IS-RC data (Supplementary Figure 1a, Table 2, Supplementary Table 7). The SSC+KHC cohorts mainly contain patients with more fragmented sleeping patterns, which would explain a reduced performance. The IS-RC has the most accurate label, minimizing the effects of erroneous scoring, which therefore leads to an increased performance. Incorporating large ensembles of different models increased mean performance slightly (Table 2).

The two most important factors that increased prediction accuracy were encoding and memory, while segment length, complexity and number of realizations were less important (Supplementary Figure 1). The effect of encoding was less

prominent in the IS-RC. Prominent factor interactions include (Supplementary Figure 2): (i) CC encoding models improve with higher complexity, whereas octave encoding models worsen; (ii) increasing segment length positively affects models with low complexity, but does not affect models with a high complexity; and (iii) adding memory improves models with an octave encoding more than models with a CC encoding. Because the IS-RC data are considered the most reliable, we decided to use these data as benchmark for model comparison. This standard improved as more scorers were added, and the model performance increased. (Fig. 1a). The different model configurations described in this section do not represent exhaustive configuration search, and future work experiments might result in improved results.

Figure 2a displays typical scoring outputs (bottom panels) obtained with a single sleep study of the IS-RC cohort in comparison to 6 scorer consensus (top panel). The model results are displayed as hypnodensity graphs, representing not only discrete sleep stage outputs, but also the probability of occurrence of each sleep state for each epoch (see definition in Data labels, scoring and fuzzy logic section). As can be seen, all models performed well, and segments of the sleep study with the lowest scorer consensus (top) are paralleled by similar sleep stage probability uncertainty, with performance closest to scoring consensus achieved by an ensemble model described below (second to top).

**Final implementation of automatic sleep scoring algorithm**. Because of model noise, potential inaccuracies and the desire to quantify uncertainty, the final implementation of our sleep scoring algorithm is an ensemble of different CC models with small variations in model parameters, such as the number of feature-maps and hidden nodes. This was achieved by randomly varying the parameters between 50 and 150% of the original values using the CC/SH/LS/LSTM as a template (this model achieved similar performance to the CC/LH/LS/LSTM while requiring significantly less computational power).

All models make errors, but as these errors occur independently of each other, the risk of not detecting and correcting errors falls with increasing model numbers. For this reason, 16 such models were trained, and at each analyzed segment both mean and variance of model estimates were calculated. As expected, the relative model variance (standardized to the average variance in a correct wakefulness prediction) is generally lower in correct predictions (Supplementary Table 3) and this can be used to inform users about uncertain/incorrect estimates. To demonstrate the effectiveness of this final implementation, the average of the models is shown alongside the distribution of 5234 ± 14 scorers on 150 epochs, a dataset provided by the AASM (AASM inter-scorer reliability (ISR) dataset, (see Datasets section

**Table 2 Performance of best models, as they are described by Supplementary Table 8, on various datasets compared to the six-scorer consensus**

| Test data | Best single model | Mean performance (%) | Best ensemble | Mean performance (%) |
|---|---|---|---|---|
| WSC | CC/SH/LS/LSTM/2 | 86.0 ± 5.0 | All CC | 86.4 ± 5.2 |
| SSC+KHC, no narcolepsy | CC/LH/SS/LSTM | 76.9 ± 11.1 | All CC | 77.0 ± 11.9 |
| SSC+KHC, narcolepsy | CC/LH/SS/LSTM | 68.8 ± 11.0 | All CC | 68.4 ± 12.2 |
| IS-RC | CC/LH/LS/LSTM/2 | 84.6 ± 4.6 | All models | 86.8 ± 4.3 |

All comparisons are on a by-epoch basis

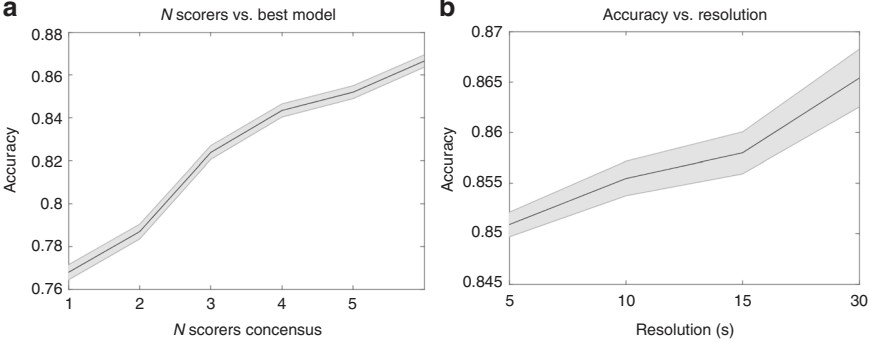

**Fig. 1** Accuracy per scorer and by time resolution. **a** The effect on scoring accuracy as golden standard is improved. Every combination of N scorers is evaluated in an unweighted manner and the mean is calculated. Accuracy is shown with mean (solid black line) and a 95% confidence interval (gray area). **b** Predictive performance of best model at different resolutions. Performance is shown as mean accuracy (solid black line) with a 95% confidence interval (gray area)

in Methods). On these epochs, the AASM ISR achieved a 90% agreement between scorers. In comparison, the model estimates reached a 95% accuracy compared to the AASM consensus (Fig. 2b). Using the model ensemble and reporting on sleep stage probabilities and inter-model variance for quality purpose constitute the core of our sleep scoring algorithm.

**Ensemble/best model performance**. Supplementary Table 2 reports on concordance for our best model, the ensemble of all CC models. Concordance is presented in a weighted and unweighted manner, between the best model estimate and scorer consensus (Table 3). Weighing of a segment was based on scorer confidence and serves to weigh down controversial segments. For each recording $i$, the epoch-specific weight $\omega_n$ and weighted accuracy $\alpha_\omega$ were calculated as:

$$\omega_n = \max_{z \in \mathcal{Z}} \left( \mathbf{P}(\mathbf{y}_n | \mathbf{x}_n)_z \right) - \ell^2_{\mathcal{Z}} \left( \mathbf{P}(\mathbf{y}_n | \mathbf{x}_n) \right), \quad (2)$$

$$\alpha_\omega^{(i)} = \frac{1}{\sum_n \omega_n} \sum_n \omega_n \cdot \left( \arg\max_{m \in \mathcal{M}} \left( \mathbf{P}_m(\widehat{\mathbf{y}}_n | \mathbf{x}_n) \right) \\ \cap \arg\max_{z \in \mathcal{Z}} \left( \mathbf{P}_z(\mathbf{y}_n | \mathbf{x}_n) \right) \right), \quad (3)$$

where $\ell^2_{\mathcal{Z}} \left( \mathbf{P}(\mathbf{y}_n | \mathbf{x}_n) \right)$ is the second most likely stage assessed by the set of scorers (experts) denoted by $\mathcal{Z}$, of the $n$th epoch in a sleep recording. As with scorers, the biggest discrepancies occurred between wake versus N1, N1 versus N2 and N2 versus N3. Additionally, the weighted performance was almost universally better than the unweighted performance, raising overall accuracy from 87 to 94%, indicating a high consensus between automatic scoring and scorers in places with high scorer confidence. An explanation for these results could be that both scorers and model are forced to make a choice between two stages when data are ambiguous. An example of this may be seen in Fig. 2a. Between 1 and 3 h, several bouts of N3 occur, although

they often do not reach the threshold for being the most likely stage. As time progresses, more evidence for N3 appears reflecting increased proportion of slow waves per epoch, and confidence increases, which finally yields "definitive" N3. This is seen in both model and scorer estimates. Choosing to present the data as hypnodensity graphs mitigates this problem. The various model estimates produce similar results, which also resemble the scorer assessment distribution, although models without memory fluctuate slightly more, and tend to place a higher probability on REM sleep in periods of wakefulness, since no contextual information is provided.

**Influences of sleep pathologies**. As seen in Table 2, the different cohorts achieve different performances. To see how much may be attributed to various pathologies, five different analyses of variance were made, with accuracy as the dependent variable, using cohort, age (grouped as age < 30, 30 ≤ age < 50 and age ≥ 50) and sex as covariates (Supplementary Table 4), investigating the effect of insomnia, OSA, restless leg syndrome (RLS), periodic leg movement index (PLMI) and T1N on accuracy of our machine learning routine versus human scoring. This was performed in the cohort mentioned above with addition of the Austrian Hypersomnia Cohort (AHC)[35]. The $p$ values obtained from paired $t$-testing for each condition were 0.75 (insomnia), $7.53 \times 10^{-4}$ (OSA), 0.13 (RLS), 0.22 (PLMI) and $1.77 \times 10^{-15}$ (T1N) respectively, indicating that only narcolepsy had a strong effect on scorer performance. Additionally, in the context of narcolepsy, cohort and age yielded $p$ values between $3.69 \times 10^{-21}$ and $2.81 \times 10^{-82}$ and between 0.62 and $6.73 \times 10^{-6}$, respectively. No significant effect of gender was ever noted. Cohort effects were expected and likely reflect local scorer performances and differences in PSG hardware and filter setups at every site. Decreased performance with age likely reflects decreased EEG amplitude, notably in N3/slow wave sleep amplitude with age[36].

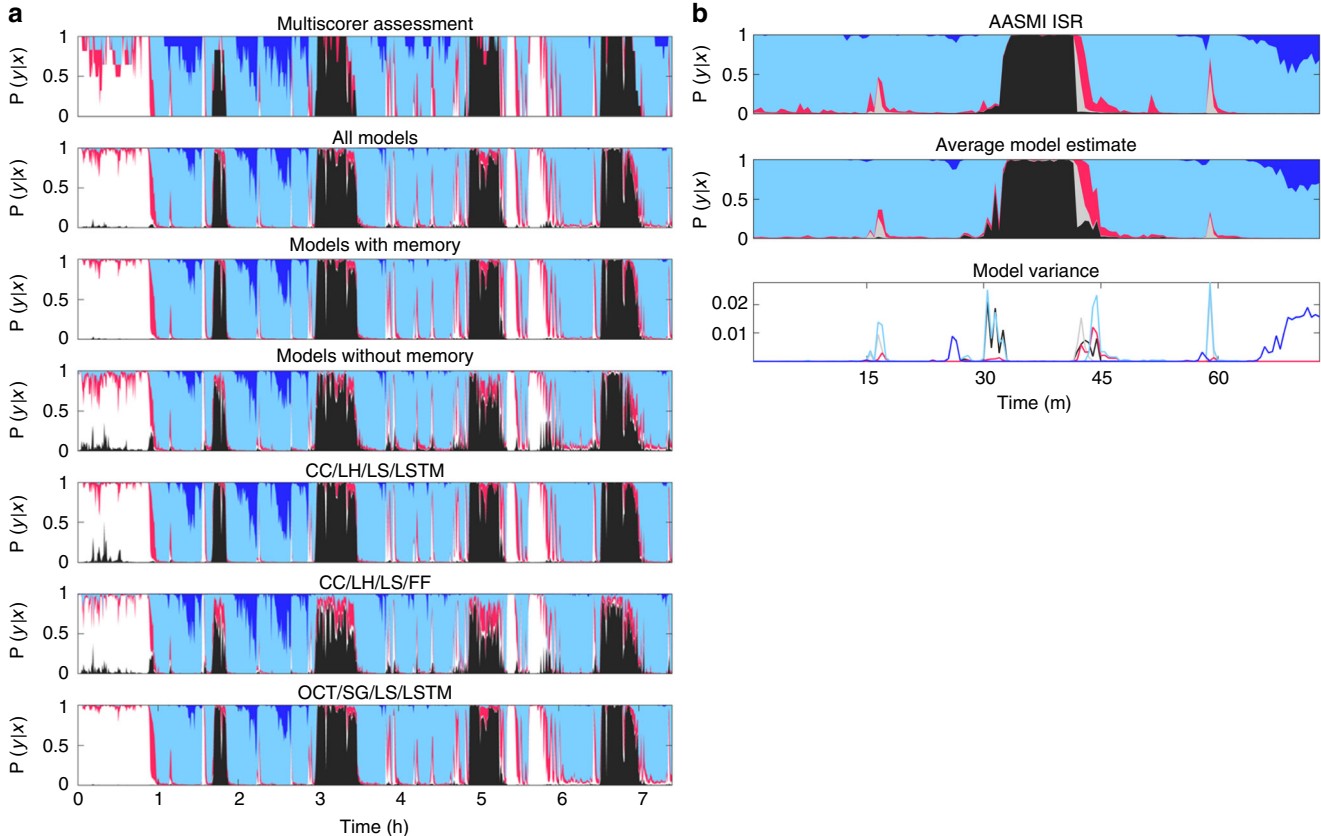

**Fig. 2** Hypnodensity example evaluated by multiple scorers and different predictive models. **a** The figure displays the hypnodensity graph. Displayed models are, in order: multiple scorer assessment (1); ensembles as described in Supplementary Table 8: All models, those with memory (LSTM) and those without memory (FF) (2–4); single models, as described in Supplementary Table 8 (5–7). OCT is octave encoding, Color codes: white, wake; red, N1; light blue, N2; dark blue, N3; black, REM. **b** The 150 epochs of a recording from the AASM ISR program are analyzed by 16 models with randomly varying parameters, using the CC/SH/LS/LSTM model as a template. These data were also evaluated by 5234 ± 14 different scorers. The distribution of these is shown on top, the average model predictions are shown in the middle, and the model variance is shown at the bottom

**Table 3 Confusion matrix displaying the relation between different targets and the ensemble estimate**

| Model Predictions | Target | | | | | |
| --- | --- | --- | --- | --- | --- | --- |
| | Wake | N1 | N2 | N3 | REM | Precision |
| Wake | 14.08% | 0.35% | 0.88% | 0.007% | 0.08% | 0.91 |
| | 16.68% | 0.15% | 0.44% | 0.003% | 0.02% | 0.96 |
| N1 | 1.13% | 1.78% | 3.00% | 0.002% | 0.36% | 0.28 |
| | 0.47% | 0.88% | 1.15% | 0% | 0.12% | 0.34 |
| N2 | 0.29% | 0.59% | 52.58% | 1.27% | 0.66% | 0.95 |
| | 0.12% | 0.25% | 56.30% | 0.34% | 0.32% | 0.98 |
| N3 | 0.002% | 0% | 2.13% | 4.87% | 0% | 0.70 |
| | 0% | 0% | 1.09% | 4.23% | 0% | 0.91 |
| REM | 0.54% | 1.17% | 0.78% | 0% | 13.45% | 0.84 |
| | 0.40% | 0.73% | 0.41% | 0% | 15.86% | 0.91 |
| Sensitivity | 0.88 | 0.46 | 0.89 | 0.79 | 0.92 | 0.87 |
| | 0.94 | 0.44 | 0.95 | 0.92 | 0.97 | 0.94 |

The targets are: top row: unweighted consensus; bottom row: weighted by the scorer agreement at each epoch. The number of analyzed epochs were 53,009 (unweighted) and 36,032 (weighted)

**Resolution of sleep stage scoring.** Epochs are evaluated with a resolution of 30 s, a historical standard that is not founded in anything physiological, and limits the analytical possibilities of a hypnogram. Consequently, it was examined to what extent the performance would change as a function of smaller resolution. Only the models using a segment size of 5 s were considered. Segments were averaged to achieve performances at 5, 10, 15 and 30 s resolutions, and the resulting performances in terms of accuracy are shown in Fig. 1b. Although the highest performance was found using a resolution of 30 s, performance dropped only slightly with decreasing window sizes.

**Construction and evaluation of a narcolepsy biomarker.** The neural networks produce outputs that depend on evidence in

the input data for or against a certain sleep stage based on features learned through training. We hypothesized that narcolepsy, a condition characterized by sleep/wake stage mixing/dissociation[37−41], would result in a greater than normal overlap between stages, an observation that was obvious when sleep stage probability were plotted in such subjects (see example in Fig. 3). Based on this result, we hypothesized that such sleep stage model outputs could be used as a biomarker for the diagnosis of narcolepsy using a standard nocturnal PSG rather than the more time-consuming MSLT.

To quantify narcolepsy-like behavior for a single recording, we generated features quantifying sleep stage mixing/dissociation. These are based on descriptive statistics and other features describing persistence of a set of new time series generated from the geometric mean of every permutation of the set of sleep stages, as obtained from the 16 CC sleep stage prediction models.

In addition to this, we also added features expected to predict narcolepsy based on prior work, such as REM sleep latency and sleep stage sequencing parameters (see "Hypnodensity as feature for the diagnosis of T1N" section in Methods for details). A recursive feature elimination (RFE) procedure[42] was performed on extracted features with average outcome putting the optimal number of relevant features at 38. An optimal selection frequency

cut-off of 0.40 (i.e., including a feature if it was selected 40% of the time) was determined using a cross-validation setup on the training data. Features are described in Supplementary Table 5 with detailed description of the 8 most important features reported in Table 4.

Final predictions were achieved by creating a separate Gaussian Predictor (GP) narcolepsy classifier from each of the sleep scoring models used in the final implementation. This was tested in seven independent datasets: a training dataset constituted of PSG from WSC[32,33], SSC[10,32], KHC[10,34], AHC[35], Jazz Clinical Trial Sample (JCTS)[43], Italian Hypersomnia Cohort (IHC)[41] and DHC; with verification in test data mostly constituted of PSG from the same cohorts and independent replication in the French Hypersomnia Cohort (FHC) and the Chinese Narcolepsy Cohort (CNC)[12] that had never been seen by the algorithm (see Supplementary Table 1). The algorithm produced values between −1 and 1, with 1 indicating a high probability of narcolepsy. A cut-off threshold between narcolepsy type 1 and "other" was set at −0.03 (red dot, Fig. 4), determined using training data, as shown in Fig. 4a. The optimal trade-off achieves both high sensitivity and specificity, which is seen to translate well onto the test data (Fig. 4b) and the never seen replication sample (Fig. 4c).

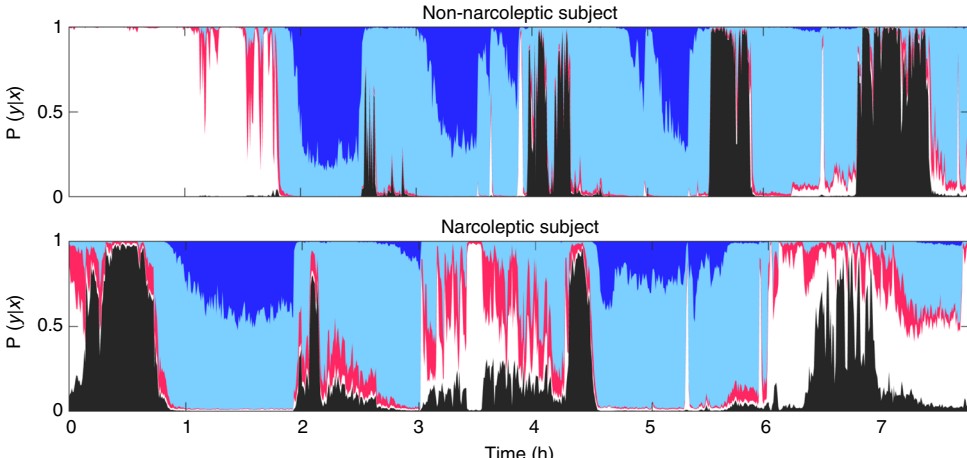

**Fig. 3** Examples of hypnodensity graph in subjects with and without narcolepsy. Hypnodensity, i.e., probability distribution per stage of sleep for a subject without narcolepsy (top) and a subject with narcolepsy (Bottom). Color codes: white, wake; red, N1; light blue, N2; dark blue, N3; black, REM

| Table 4 Descriptions of the 8 most frequently selected features | | |
|---|---|---|
| **Number** | **Relative selection frequency** | **Description** |
| 1 | 1 | The time taken before 5% of the sum of the product between W, N2 and REM, calculated at every epoch, has accumulated, weighed by the total amount of this sum. This feature expresses the known sleep stage dissociation and altered sleep timing. |
| 2 | 0.91 | The number of nightly SOREMPS appearing throughout the recording. |
| 3 | 0.82 | The time taken before 50% of the wakefulness in a recording has accumulated, weighed by the total amount of wakefulness. |
| 4 | 0.82 | REM 6 The Shannon entropy of the REM sleep stage distribution. This expresses the amount of information held in a signal, or in this case, how many different values the REM sleep stage distribution obtains—how consolidated phases of REM are when the stage appears. |
| 5 | 0.68 | The maximum probability of wakefulness obtained in a recording. |
| 6 | 0.68 | The maximum value obtained of the product between the N2 and REM probability in a recording. |
| 7 | 0.68 | The time taken before 30% of the sum of the product between W and N2, calculated at every epoch, has accumulated, weighed by the total amount of this sum. |
| 8 | 0.64 | The time taken before 10% of the sum of the product between W and N1, calculated at every epoch, has accumulated, weighed by the total amount of this sum. |

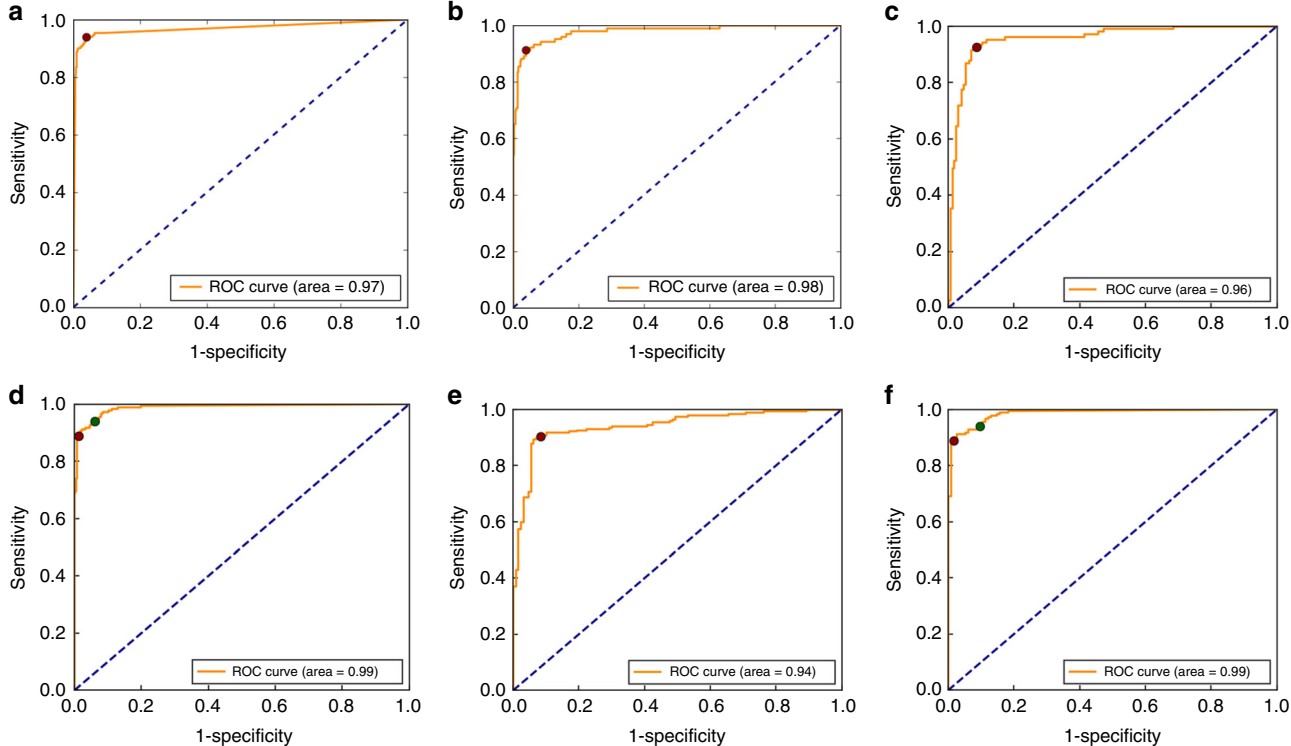

**Fig. 4** Diagnostic receiver operating characteristics curves. Diagnostic receiver operating characteristics (ROC) curves, displaying the trade-offs between sensitivity and specificity for our narcolepsy biomarker for **a** training sample, **b** testing sample, **c** replication sample and **e** high pretest sample. **d**–**f** Adding HLA to model vastly increases specificity. Cut-off thresholds are presented for models with (red dot) and without HLA (green dot)

In the training data, a sensitivity of 94% and specificity of 96% was achieved, and in the testing data a sensitivity of 91% and specificity of 96% was achieved, while the sensitivity and specificity for the replication sample was 93 and 91%, respectively. When human leukocyte antigen (HLA) was added to this model (Fig. 4d–f), the sensitivity became 90% and the specificity rose to 99%, and an updated cut-off threshold of −0.53 was determined (green dot, Fig. 4d–f). Furthermore, in the high pretest sample we obtained a sensitivity and specificity of 90 and 92%, which rose to 90 and 98% when adding HLA. More descriptive statistics including 95% confidence intervals are found in Supplementary Table 6.

## Discussion

In recent years, machine learning has been used to solve similar or more complex problems, such as labeling images, understanding speech and translating language, and have seen advancement to the point where humans are now sometimes outperformed[21–23], while also showing promising results in various medical fields[24–29]. Automatic classification of sleep stages using automatic algorithms is not novel[44,45], but only recently has this type of machine learning been applied and the effectiveness has only been demonstrated in a small numbers of sleep studies[46–49]. Because PSGs contain large amounts of manually annotated "gold standard" data, we hypothesized this method would be ideal to automatize sleep scoring. We have shown that machine learning can be used to score sleep stages in PSGs with high accuracy in multiple physical locations in various recording environments, using different protocols and hardware/software configurations, and in subjects with and without various sleep disorders.

After testing various machine learning algorithms with and without memory and specific encodings, we found increased robustness using a consensus of multiple algorithms in our prediction. The main reason for this is likely the sensitivity of each algorithm to particular aspects of each individual recording, resulting in increased or decreased predictability. Supplementary Figure 1b displays the correlations between different models. Models that incorporate an ensemble of different models generally have a higher overall correlation coefficient than singular models, and since individual models achieve similar performances, it stands to reason that these would achieve the highest performance. One potential source for this variability was, in addition to the stochastic nature of the training, the fact recordings were conducted in different laboratories that were using different hardware and filters, and had PSGs scored by technicians of various abilities. Another contributor was the presence of sleep pathologies in the dataset that could influence machine learning. Of the pathologies tested, only narcolepsy had a very significant effect on the correspondence between manual and machine learning methods ($p = 1.77 \times 10^{-15}$ vs $p = 7.53 \times 10^{-4}$ for sleep apnea for example) (Supplementary Tables 4 and 7). This was not surprising as the pathology is characterized by unusual sleep stage transitions, for example, transitions from wake to REM sleep, which may make human or machine learning staging more difficult. This result suggests that reporting inter-model variations in accuracy for each specific patient has value in flagging unusual sleep pathologies, so this metric is also reported by our detector.

Unlike previous attempts using automatic detector validations, we were able to include 70 subjects scored by 6 technicians in different laboratories (the IS-RC cohort)[31] to independently validate our best automatic scoring consensus algorithm. This allowed us to estimate the performance at 87% in comparison to the performance of a consensus score for every epoch among six expert technicians (ultimate gold standard) (Table 1). Including more scorers produces a better gold standard, and as Fig. 1a

indicates, the model accuracy also increases with more scorers. Naturally, extrapolating from this should be done with caution; however, it is reasonable to assume that the accuracy would continue to increase with increased scorers. In comparison, performance of any individual scorer ranges from 74 to 85% when compared to the same six-scorer gold standard, keeping in mind this performance is artificially inflated since the same scorers evaluated are included in the gold standard (unbiased performance of any scorer versus consensus of remaining 5 scorers range from 69 to 80%. The best model achieves 87% accuracy using 5 scorers (Fig. 1a and Table 1), and is statistically higher than all scorers. As with human scorers, the biggest discrepancies in machine learning determination of sleep stages occurred between wake versus N1, N1 versus N2 and N2 versus N3. This is logical as these particular sleep stage transitions are part of a continuum, artificially defined and subjective. To give an example: an epoch comprised of 18% slow wave activity is considered N2 while an epoch comprised of 20% slow wave activity qualifies as N3. Overall, data indicate that our machine learning algorithm performs better than individual scorers, as typically used in clinical practice, or similar to the best of 5 scorers in comparison to a combination of 5 experts scoring each epoch by consensus. It is also able to score at higher resolution, i.e., 5 s, making it unnecessary to score sleep stages by 30 s epochs, an outdated rule dating from the time sleep was scored on paper. Although the data sample used for multi-scorer validation contained only female subjects, the scoring accuracy of our model was not seen to be affected by gender (Supplementary Table 3) in another analysis.

Using our models, and considering how typical T1N behaved in our sleep stage machine learning routines, we extracted features that could be useful to diagnose this condition. T1N is characterized by the loss of hypocretin-producing cells in the hypothalamus[3] and can be best diagnosed by measuring hypocretin levels in the CSF[11], a procedure that requires a lumbar puncture, a rarely performed procedure in the United States. At the symptomatic level, T1N is characterized by sleepiness, cataplexy (episodes of muscle weakness during wakefulness triggered by emotions) and numerous symptoms reflecting poor nocturnal sleep (insomnia) and symptoms of "dissociated REM sleep". Dissociated REM sleep is reflected by the presence of unusual states of consciousness where REM sleep is intermingled with wakefulness, producing disturbing reports of dreams that interrupt wakefulness and seem real (dream-like hallucinations), or episodes where the sleeper is awake but paralyzed as in normal REM sleep (sleep paralysis). The current gold standard for T1N diagnosis is the presence of cataplexy and a positive MSLT. In a recent large study of the MSLT, specificity and sensitivity for T1N was 98.6% and 92.9% in comparing T1N versus controls, and 71.2% and 93.4% in comparing T1N versus other hypersomnia cases (high pretest probability cohort)[10].

Table 4 and Supplementary Table 5 reveal features found in nocturnal PSGs that discriminate type 1 narcoleptics and non-narcoleptics. One of the most prominent features, short latency REM sleep, bears great resemblance to the REM sleep latency, which is already used clinically to diagnose narcolepsy, although in this case it is calculated using fuzzy logic and thus represent a latency where accumulated sleep is suggestive of a high probability of REM sleep having occurred (as opposed to a discrete REM latency scored by a technician). A short REM latency during nocturnal PSG (typically 15 min) has recently been shown to be extremely specific (99%) and moderately sensitive (40–50%) for T1N[10,50]. The remaining selected features also describe a generally altered sleep architecture, particularly between REM sleep, light sleep and wake, aspects of narcolepsy already known and thus reinforcing their validity as biomarkers.

For example, the primary feature as determined by the RFE algorithm was the time taken until 5% of the accumulated sum of the probability products between stages W, N2 and REM had been reached (see also Table 4), which reflects the uncertainty between wakefulness, REM and N2 sleep at the beginning of the night. Specifically, for the $n^{th}$ epoch, the model will output probabilities for each sleep stage, and the proto-feature $\Phi_n$ is calculated as

$$\Phi_n = p(W) \times p(N2) + p(W) \times p(REM) + p(N2) \times p(REM).$$
$$(4)$$

The feature value is then calculated as the time it takes in minutes for the accumulated sum of $\Phi_n$ to reach 5% of the total sum $\sum_n \Phi_n$. Since each of probability product in $\Phi_n$ reflects the staging uncertainty between each sleep stage pair, $\Phi_n$ alone reflects the general sleep stage uncertainty for that specific epoch as predicted by the model. A very high value will be attained for epoch $n$ if the probabilities for N2, W and REM are equally probable with probabilities for the remaining sleep stages being low or close to zero. A PSG with a high staging uncertainty between sleep and wake early in the night would reach the 5% threshold rapidly.

Using these features, we were able to determine an optimal cut-off that discriminated narcolepsy from controls and any other patients with as high specificity and sensitivity as the MSLT (Supplementary Table 6), notably when HLA typing is added. This is true for both the test and the never seen replication samples. Although we do observe a small drop in specificity in the replication sample, the efficacy of the detector was also tested in the context of naive patients with hypersomnia (high pretest probability sample), and performance found to be similar to the MSLT.

MSLT testing requires that patients spend an entire night and day in a sleep laboratory. The use of this novel biomarker could reduce time spent to a standard 8 h night recording, as done for the screening of other sleep pathologies (e.g., OSA), allowing improved recognition of T1N cases at a fraction of the cost. A positive predictive value could also be provided depending on the nature of the sample and known narcolepsy prevalence (low in general population screening, intermediary in overall clinic population sample and high in hypersomnia cohorts). It also opens the possibility of using home sleep recordings for diagnosing narcolepsy. In this direction, because of the probabilistic and automatic nature of our biomarker, estimates from more than one night could be automatically analyzed and combined over time, ensuring improved prediction. However, it is important to note that this algorithm will not replace the MSLT in the ability to predict excessive daytime sleepiness through the measure of mean sleep latency across daytime naps, which is an important characteristic of other hypersomnias.

In conclusion, models which classify sleep by assigning a membership function to each of five different stages of sleep for each analyzed segment were produced, and factors contributing to the performance were analyzed. The models were evaluated on different cohorts, one of which contained 70 subjects scored by 6 different sleep scoring technicians, allowing for inter-scorer reliability assessments. The most successful model, consisting of an ensemble of different models, achieved an accuracy of 87% on this dataset, and was statistically better performing than any individual scorer. It was also able to score sleep stages with high accuracy at lower time resolution (5 s), rendering the need for scoring per 30 s epoch obsolete. When predictions were weighted by the scorer agreement, performance rose to 95%, indicating a high consensus between the model and human scorers in areas of high scorer agreement. A final implementation was made using

an ensemble with small variations of the best single model. This allowed for better predictions, while also providing a measure of uncertainty in an estimate.

When the staging data were presented as hypnodensity distributions, the model conveyed more information about the subject than through a hypnogram alone. This led to the creation of a biomarker for narcolepsy that achieved similar performance to the current clinical gold standard, the MSLT, but only requires a single sleep study. If increased specificity is needed, for example, in large-scale screening, HLA or additional genetic typing brings specificity above 99% without loss of sensitivity. This presents an option for robust, consistent, inexpensive and simpler diagnosis of subjects who may have narcolepsy, as such tests may also be carried out in a home environment.

This study shows how hypnodensity graphs can be created automatically from raw sleep study data, and how the resulting interpretable features can be used to generate a diagnosis probability for T1N. Another approach would be to classify narcolepsy directly from the neural network by optimizing the performance not only for sleep staging, but also for direct diagnosis by adding an additional softmax output, thereby creating a multitask classifier. This approach could lead to better predictions, since features are not then limited to by a designer imagination. A drawback of this approach is that features would no longer be as interpretable and meaningful to clinicians. If meaning could be extracted from these neural network generated features, this might open the door to a single universal sleep analysis model, covering multiple diseases. Development of such a model would require adding more subjects with narcolepsy and other conditions to the pool of training data.

## Methods

**Datasets.** The success of machine learning depends on the size and quality of the data on which the model is trained and evaluated[51,52]. We used a large dataset comprised of several thousand sleep studies to train, validate and test/replicate our models. To ensure significant heterogeneity, data came from 10 different cohorts recorded at 12 sleep centers across 3 continents: SSC[10,32], WSC[32,33], IS-RC[31], JCTS[43], KHC[10,34], AHC[35], IHC[41], DHC[53], FHC and CNC[12]. Institutional review boards approved the study and informed consent was obtained from all participants. Technicians trained in sleep scoring manually labeled all sleep studies. Figure 5a–c summarizes the overall design of the study for sleep stage scoring and narcolepsy biomarker development. Supplementary Table 1 provides a summary of the size of each cohort and how it was used. In the narcolepsy biomarker aspect of the study, PSGs from T1N and other patients were split across most datasets to ensure heterogeneity in both the training and testing datasets. For this analysis, a few recordings with poor quality sleep studies, i.e., missing critical channels, with additional sensors or with a too short sleep duration (≤2 h) were excluded. A "never seen" subset cohort that included French and Chinese subjects (FHC and CNC) was also tested. Below is a brief description of each dataset.

**Population-based Wisconsin Sleep Cohort.** This cohort is a longitudinal study of state agency employees aged 37–82 years from Wisconsin, and it approximates a population-based sample (see Supplementary Table 1 for age at study) except for

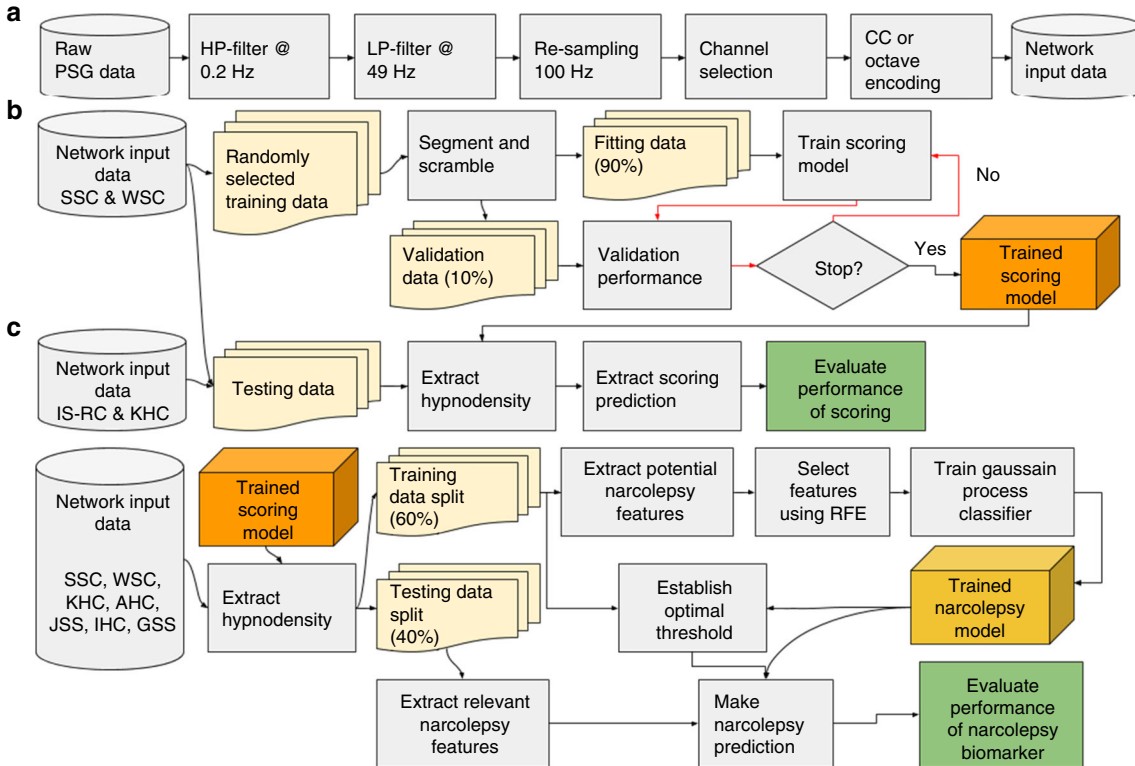

**Fig. 5** Overall design of the study. **a** Pre-processing steps taken to achieve the format of data as it is used in the neural networks. One of the 5 channels is first high-pass filtered with a cut-off at 0.2 Hz, then low-pass filtered with a cut-off at 49 Hz followed by a re-sampling to 100 Hz to ensure data homogeneity. In the case of EEG signals, a channel selection is employed to choose the channel with the least noise. The data are then encoded using either the CC or the octave encoding. **b** Steps taken to produce and test the automatic scoring algorithm. A part of the SSC[10, 32] and WSC[32, 33] is randomly selected, as described in Supplementary Table 1. These data are then segmented in 5 min segments and scrambled with segments from other subjects to increase batch similarity during training. A neural network is then trained until convergence (evaluated using a separate validation sample). Once trained, the networks are tested on a separate part of the SSC and WSC along with data from the IS-RC[31] and KHC[10, 34]. **c** Steps taken to produce and test the narcolepsy detector. Hypnodensities are extracted from data, as described in Supplementary Table 1. These data are separated into a training (60%) and a testing (40%) split. From the training split, 481 potentially relevant features, as described in Supplementary Table 9, are extracted from each hypnodensity. The prominent features are maintained using a recursive selection algorithm, and from these features a GP classifier is created. From the testing split, the same relevant features are extracted, and the GP classifier is evaluated

the fact they are generally more overweight[33]. The study is ongoing, and dates to 1988. The 2167 PSGs in 1086 subjects were used for training, while 286 randomly selected PSGs were used for validation testing of the sleep stage scoring algorithm and narcolepsy biomarker training. Approximately 25% of the population have an Apnea Hypopnea Index (AHI) above 15/h and 40% have a PLMI above 15/h. A detailed description of the sample can be found in Young et al.[33] and Moore et al.[32]. The sample does not contain any T1N patients, and the three subjects with possible T1N were removed[54].

**Patient-based Stanford Sleep Cohort.** PSGs from this cohort were recorded at the Stanford Sleep Clinic dating back to 1999, and represent sleep disorder patients aged 18–91 years visiting the clinic (see Supplementary Table 1 for age at study). The cohort contains thousands of PSG recordings, but for this study we used 894 diagnostic (no positive airway pressure) recordings in independent patients that have been used in prior studies[30]. This subset contains patients with a range of different diagnoses including: sleep disordered breathing (607), insomnia (141), REM sleep behavior disorder (4), restless legs syndrome (23), T1N (25), delayed sleep phase syndrome (14) and other conditions (39). Description of the subsample can be found in Andlauer et al.[10] and Moore et al.[32]. Approximately 30% of subjects have an AHI above 15/h, or a PLMI above 15/h. The 617 randomly selected subjects were used for training the neural networks, while 277 randomly selected PSGs were kept for validation testing of the sleep stage scoring algorithm. These 277 subjects were also used for training the narcolepsy biomarker algorithm. The sample contains PSGs of 25 independent untreated subjects with T1N (12 with low CSF hypocretin-1, the others with clear cataplexy). A total of 26 subjects were removed from the study—4 due to poor data quality, and the rest because of medication use.

**Patient-based Korean Hypersomnia Cohort.** The Korean Hypersomnia Cohort is a high pretest probability sample for narcolepsy. It includes 160 patients with a primary complaint of excessive daytime sleepiness (see Supplementary Table 1 for age at study). These PSGs were used for testing the sleep scoring algorithm and for training the narcolepsy biomarker algorithm. No data were used for training the sleep scoring algorithm. Detailed description of the sample can be found in Hong et al.[34] and Andlauer et al.[10]. The sample contains PSGs of 66 independent untreated subjects with T1N and clear cataplexy. Two subjects were removed from the narcolepsy biomarker study because of poor data quality.

**Patient-based Austrian Hypersomnia Cohort.** Patients in this cohort were examined at the Innsbruck Medical University in Austria as described in Frauscher et al.[35]. The AHC contains 118 PSGs in 86 high pretest probability patients for narcolepsy (see Supplementary Table 1 for details). The 42 patients (81 studies) are clear T1N with cataplexy cases, with all but 3 having a positive MSLT (these three subjects had a MSL >8 min but multiple SOREMPs). The rest of the sample has idiopathic hypersomnia and type 2 narcolepsy. Four patients have an AHI >15/h and 25 had a PLMI >15/h. Almost all subjects had two sleep recordings performed, which were kept together such that no two recordings from the same subject were split between training and testing partitions.

**Patient-based Inter-scorer Reliability Cohort.** As Rosenberg and Van Hout[16] have shown, variation between individual scorers can sometimes be large, leading to an imprecise gold standard. To quantify this, and to establish a more accurate gold standard, 10 scorers from 5 different institutions, University of Pennsylvania, St. Luke's Hospital, University of Wisconsin at Madison, Harvard University and Stanford University, analyzed the same 70 full-night PSGs. For this study, scoring data from University of Pennsylvania, St. Luke's and Stanford were used. All subjects are female (see Supplementary Table 1 for details). This allowed for a much more precise gold standard, and the inter-scorer reliability could be quantified for a dataset, which could also be examined by automatic scoring algorithms. Detailed description of the sample can be found in Kuna et al.[31] and Malhotra and Avidan[6]. The sample does not contain any T1N patients.

**The Jazz Clinical Trial Sample.** This sample includes 7 baseline sleep PSGs from 5 sites taken from a clinical trial study of sodium oxybate in narcolepsy (SXB15 with 45 sites in Canada, United States, and Switzerland) conducted by Orphan Medical, now named Jazz Pharmaceuticals. The few patients included are those with clear and frequent cataplexy (a requirement of the trial) who had no stimulant or antidepressant treatment at baseline[43]. All seven subjects in this sample were used exclusively for training the narcolepsy biomarker algorithm.

**Patient-based Italian Hypersomnia Cohort.** Patients in this high pretest probability cohort (see Supplementary Table 1 for demographics) were examined at the IRCCS, Istituto delle Scienze Neurologiche ASL di Bologna in Italy as described in Pizza et al.[41]. The IHC contains 70 T1N patients (58% male, 29.5 ± 1.9 years old), with either documented low CSF hypocretin levels (59 cases, all but 2 HLA-DQB1*06:02 positive) or clear cataplexy, positive MSLTs and HLA positivity (11 subjects). As non-T1N cases with unexplained daytime somnolence, the cohort includes 77 other patients: 19 with idiopathic hypersomnia, 7 with type 2

narcolepsy and normal CSF hypocretin-1, 48 with a subjective complaint of excessive daytime sleepiness not confirmed by MSLT and 3 with secondary hypersomnia. Subjects in this cohort were used for training ($n = 87$) and testing ($n = 61$) the narcolepsy biomarker algorithm.

**Patient-based Danish Hypersomnia Cohort.** Patients in this cohort were examined at the Rigshospitalet, Glostrup, Denmark, as described in Christensen et al.[53]. The DHC contains 79 PSGs in controls and patients (see Supplementary Table 1 for details). Based on PSG, multiple sleep latency test and cerebrospinal fluid hypocretin-1 measures, the cohort includes healthy controls (19 subjects), patients with other sleep disorders and excessive daytime sleepiness (20 patients with CSF hypocretin-1 ≥110 pg/ml), narcolepsy type 2 (22 patients with CSF hypocretin-1 ≥110 pg/ml), and T1N (28 patients with CSF hypocretin-1 ≤110 pg/ml). All 79 subjects in this cohort were used exclusively for training the narcolepsy biomarker algorithm.

**Patient-based French Hypersomnia Cohort.** This cohort consists of 122 individual PSGs recorded at the Sleep-Wake Disorders Center, Department of Neurology, Gui-de-Chauliac Hospital, CHU Montpellier, France (see Supplementary Table 1 for demographics). The FHC contains 63 subjects with T1N (all but two tested with CSF hypocretin-1 ≤110 pg/ml, five below 18 years old, 55 tested for HLA, all positive for HLA-DQB1*06:02) and 22 narcolepsy type 2 (19 with CSF hypocretin-1 >200 pg/ml, and three subjects with CSF hypocretin-1 between 110 and 200 pg/ml, three HLA positive). The remaining 36 subjects are controls (15 tested for HLA, two with DQB1*06:02) without other symptoms of hypersomnia. The FHC was used as data for the replication study of the narcolepsy biomarker algorithm.

**Patient-based Chinese Narcolepsy Cohort.** This cohort contains 199 individual PSGs recorded (see Supplementary Table 1 for demographics). The CNC contains 67 subjects diagnosed with T1N exhibiting clear-cut cataplexy (55 tested HLA-DQB1*06:02 positive), while the remaining 132 subjects are randomly selected population controls (15 HLA-DQB1*06:02 positive, 34 HLA negative, remaining unknown)[12]. Together with the FHC, the CNC was used as data for the replication study of the narcolepsy biomarker algorithm.

**American Academy of Sleep Medicine Sleep Study.** The AASM ISR dataset is composed of a single control sleep study of 150 30 s epochs that was scored by 5234 ± 14 experienced sleep technologists for quality control purposes. Design of this dataset is described in Rosenberg and Van Hout[16].

**Data labels, scoring and fuzzy logic.** Sleep stages were scored by PSG-trained technicians using established scoring rules, as described in the AASM Scoring Manual[7]. In doing so, technicians assign each epoch with a discrete value. With a probabilistic model, like the one proposed in this study, a relationship to one of the fuzzy sets is inferred based on thousands of training examples labeled by many different scoring technicians.

The hypnodensity graph refers to the probability distribution over each possible stage for each epoch, as seen in Fig. 2a, b. This allows more information to be conveyed, since every epoch of sleep within the same stage is not identical. For comparison with the gold standard, however, a discrete value must be assigned from the model output as:

$$\hat{y} = \text{argmax}_{\mathbf{y}_i} \sum_i^N \mathbf{P}_i(\mathbf{y}_i | \mathbf{x}_i), \tag{5}$$

where $\mathbf{P}_i(\mathbf{y}_i | \mathbf{x}_i)$ is a vector with the estimated probabilities for each sleep stage in the $i^{\text{th}}$ segment, $N$ is the number of segments an epoch is divided into and $\hat{y}$ is the estimated label.

Sleep scoring technicians score sleep in 30 s epochs, based on what stage they assess is represented in the majority of the epoch—a relic of when recordings were done on paper. This means that when multiple sleep stages are represented, more than half of the epoch may not match the assigned label. This is evident in the fact that the label accuracy decreases near transition epochs[20]. One solution to this problem is to remove transitional regions to purify each class. However, this has the disadvantage of under-sampling transitional stages, such as N1, and removes the context of quickly changing stages, as is found in a sudden arousal. It has been demonstrated that the negative effects of imperfect "noisy" labels may be mitigated if a large enough training dataset is incorporated and the model is robust to overfitting[41]. This also assumes that the noise is randomly distributed with an accurate mean—a bias cannot be canceled out, regardless of the amount of training data. For these reasons, all data including those containing sleep transitions were included. Biases were evaluated by incorporating data from several different scoring experts cohorts and types of subjects.

To ensure quick convergence, while also allowing for long-term dependencies in memory-based models, the data were broken up in 5 min blocks and shuffled to minimize the shift in covariates during training caused by differences between subjects. To quantify the importance of segment sizes, both 5 s and 15 s windows were also tested.

**Data selection and pre-processing.** A full-night PSG involves recording many different channels, some of which are not necessary for sleep scoring[55]. In this study, EEG, C3 or C4, and O1 or O2, chin EMG and the left and right EOG channels were used, with reference to the contralateral mastoid. Poor electrode connections are common when performing a PSG analysis. This can lead to a noisy recording, rendering it useless. To determine whether right or left EEG channels were used, the noise of each was quantified by dividing the EEG data in 5 min segments, and extracting the Hjorth parameters[56]. These were then log-transformed, averaged and compared with a previously established multivariate distribution, based on the WSC[32,33] and SSC[10,32] training data. The channel with lowest Mahalanobis distance[57] to this distribution was selected. The log transformation has the advantage of making flat signals/disconnects as uncommon as very noisy signals, in turn making them less likely to be selected. To minimize heterogeneity across recordings, and at the same time reducing the size of the data, all channels were down-sampled to 100 Hz. Additionally, all channels were filtered with a fifth-order two-direction infinite impulse response (IIR) high-pass filter with cut-off frequency of 0.2 Hz and a fifth-order two-direction IIR low-pass filter with cut-off frequency of 49 Hz. The EMG signal contains frequencies well above 49 Hz, but since much data had been down-sampled to 100 Hz in the WSC, this cut-off was selected for all cohorts. All steps of the pre-processing are illustrated in Fig. 5a.

**Convolutional and recurrent neural networks.** Convolutional neural networks (CNNs) are a class of deep learning models first developed to solve computer vision problems[30]. A CNN is a supervised classification model in which a low level, such as an image, is transformed through a network of filters and sub-sampling layers. Each layer of filters produces a set of features from the previous layer, and as more layers are stacked, more complex features are generated. This network is coupled with a general-purpose learning algorithm, resulting in features produced by the model reflecting latent properties of the data rather than the imagination of the designer. This property places fewer constrictions on the model by allowing more flexibility, and hence the predictive power of the model will increase as more data are observed. This is facilitated by the large number of parameters in such a model, but may also necessitate a large amount of training data. Sleep stage scoring involves a classification of a discrete time series, in which adjacent segments are correlated. Models that incorporate memory may take advantage of this and may lead to better overall performance by evening out fluctuations. However, these fluctuations may be the defining trait or anomaly of some underlying pathology (such as narcolepsy, a pathology well known to involve abnormal sleep stages transitions), present in only a fraction of subjects, and perhaps absent in the training data. This can be thought of similarly to a person with a speech impediment: the contextual information will ease the understanding, but knowing only the output, this might also hide the fact that the person has such a speech impediment. To analyze the importance of this, models with and without memory were analyzed. Memory can be added to such a model by introducing recurrent connections in the final layers of the model. This turns the model into a recurrent neural network (RNN). Classical RNNs had the problem of vanishing or exploding gradients, which meant that optimization was very difficult. This problem was solved by changing the configuration of the simple hidden node into a LSTM cell[58]. Models without this memory are referred to as FF models. A more in-depth explanation of CNNs including application areas can be found in the review article on deep learning by LeCun et al.[30] and the deep learning textbook by Goodfellow et al.[59]. For a more general introduction to machine learning concepts, see the textbook by Bishop[60].

**Data input and transformations.** Biophysical signals, such as those found in a PSG, inherently have a low signal to noise ratio, the degree of which varies between subjects, and hence learning robust features from these signals may be difficult. To circumvent this, two representations of the data that could minimize these effects were selected. An example of each decomposition is shown in Fig. 6a.

Octave encoding maintains all information in the signal, and enriches it by repeatedly removing the top half of the bandwidth (i.e., cut-off frequencies of 49, 25, 12.5, 6.25 and 3.125 Hz) using a series of low-pass filters, yielding a total of 5 new channels for each original channel. At no point is a high-pass filter applied. Instead, the high frequency information may be obtained by subtracting lower frequency channels—an association the neural networks can make, given their universal approximator properties[61]. After filtration, each new channel is scaled to the 95th percentile and log modulus transformed:

$$\mathbf{x}_{\text{scaled}} = \text{sign}(\mathbf{x}) \cdot \log\left(\frac{|\mathbf{x}|}{P_{95}(\mathbf{x})} + 1\right). \tag{6}$$

The initial scaling places 95% of the data between −1 and 1, a range in which the log modulus is close to linear. Very large values, such as those found in particularly noisy areas, are attenuated greatly. Some recordings are noisy, making the 95th percentile significantly higher than what the physiology reflects. Therefore, instead of selecting the 95th percentile from the entire recording, the recording is separated into 50% overlapping 90 min segments, from which the 95th percentile is extracted. The mode of these values is then used as a scaling reference. In general, scaling and normalization is important to ensure quick convergence as well as generalization in neural networks. The decomposition is done in the same way on every channel, resulting in 25 new channels in total.

CC encoding, using a CC function, underlying periodicities in the data are revealed while noise is attenuated. White noise is by definition uncorrelated; its autocorrelation function is zero everywhere except lag zero. It is this property that is utilized, even though noise cannot always be modeled as such. PSG signals are often obscured by undesired noise that is uncorrelated with other aspects of the signals. An example CC between a signal segment and an extended version of the same signal segment is shown in Supplementary Figure 5. Choosing the CC in this manner over a standard autocorrelation function serves two purposes: the slow frequencies are expressed better, since there is always full overlap between the two signals (some of this can be adjusted with the normal autocorrelation function using an unbiased estimate); and the change in fluctuations over time within a segment is expressed, making the function reflect aspects of stationarity. Because this is the CC between a signal and an extended version of itself, the zero lag represents the power of that segment, as is the case in an autocorrelation function.

Frequency content with a time resolution may also be expressed using time-frequency decompositions, such as spectrograms or scalograms; however, one of the key properties of a CNN is the ability to detect distinct features anywhere in an input, given its property of equivariance[62]. A CC function reveals an underlying set of frequencies as an oscillation pattern, as opposed to a spectrogram, where frequencies are displayed as small streaks or spots in specific locations, corresponding to frequencies at specific times. The length and size of each CC reflects the expected frequency content and the limit of quasi-stationarity (i.e., how quickly the frequency content is expected to change).

The EOG signal reveals information about eye movements such as REMs, and to some extent EEG activity[6,7]. In the case of the EOG signal, the relative phase between the two channels is of great importance to determine synchronized eye movements, and hence a CC of opposite channels (i.e., either the extended or zero padded signal is replaced with the opposite channel) is also included. The slowest eye movements happen over the course of several seconds[6,7], and hence a segment length of 4 s was selected for the correlation functions. To maintain resolution flexibility with the EEG, an overlap of 3.75 s was chosen.

In the case of the EMG signal, the main concern is the signal amplitude and the temporal resolution, not the actual frequencies. As no relevant low-frequency content is expected, a segment length of 0.4 s and an overlap of 0.25 s was selected.

As with the octave encoding, the data are scaled, although only within segments:

$$D_i = \frac{\gamma_{\mathbf{x}_i \mathbf{y}_i} \cdot \log\left(1 + \max\left(\left|\gamma_{\mathbf{x}_i \mathbf{y}_i}\right|\right)\right)}{\max\left(\left|\gamma_{\mathbf{x}_i \mathbf{y}_i}\right|\right)}, \tag{7}$$

where $D_i$ is the scaled correlation function and $\gamma_{\mathbf{x}_i \mathbf{y}_i}$ is the unscaled correlation function.

**Architectures of applied CNN models.** The architecture of a CNN typically reflects the complexity of the problem that is being solved and how much training data are available, as a complex model has more parameters than a simple model, and is therefore more likely to over-fit. However, much of this may be solved using proper regularization. Another restriction is the resources required to train a model —deep and complex models require far more operations and will therefore take longer to train and operate. In this study, no exhaustive hyper-parameter optimization was carried out. The applied architectures were chosen on the basis of other published models[63]. Since the models utilized three separate modalities (EEG, EOG and EMG), three separate sub-networks were constructed. These were followed by fully connected layers combining the inputs from each sub-network, which were passed onto a softmax output (Fig. 6b, Supplementary Figure 3). Models that utilize memory have fully connected hidden units replaced with LSTM cells and recurrent connections added between successive segments. Networks of two different sizes are evaluated to quantify the effect of increasing complexity.

**Training of CNN models.** Training the models involves optimizing parameters to minimize a loss function evaluated across a training dataset. The loss function was defined as the cross-entropy with L2 regularization:

$$L(\boldsymbol{\omega}) = \frac{1}{N}\sum_{i=1}^{N} H(\mathbf{y}_i, \widehat{\mathbf{y}}_i) + L2 = \frac{1}{N}\sum_{i=1}^{N} \mathbf{y}_i \log \widehat{\mathbf{y}}_i + (1 - \mathbf{y}_i)\log(1 - \widehat{\mathbf{y}}_i) + \lambda \|\boldsymbol{\omega}\|_2^2, \tag{8}$$

where $\mathbf{y}_i$ is the true class label of the $i^{\text{th}}$ window, $\widehat{\mathbf{y}}_i$ is the estimated probability of the $i^{\text{th}}$ window, $\boldsymbol{\omega}$ is the parameter to be updated and $\lambda$ is the weight decay parameter set at 0.00001. The model parameters were initialized with $N(0, 0.01)$, and trained until convergence using stochastic gradient decent with momentum[64]. Weight updates were done as: $\boldsymbol{\omega}_{t+1} = \boldsymbol{\omega}_t + \eta \mathbf{v}_{t+1}$ with $\mathbf{v}_{t+1} = \alpha \mathbf{v}_t - \frac{\delta E}{\delta \omega_t}$ where $\alpha$ is the momentum set at 0.9, $\mathbf{v}_t$ is the learning velocity, initialized at 0, and $\eta$ is the learning rate, initially set at 0.005. The learning rate was gradually reduced with an exponential decay $\eta = \eta_0 \cdot e^{-t/\tau}$ where $t$ is the number of updates and $\tau$ is a time constant, here set to 12,000.

Overfitting was avoided using a number of regularization techniques, including batch normalization[65], weight decay[66] and early stopping[67]. Early stopping is accomplished by scheduling validation after every 50th training batch. This is done by setting aside 10% of the training data. Training is stopped if the validation

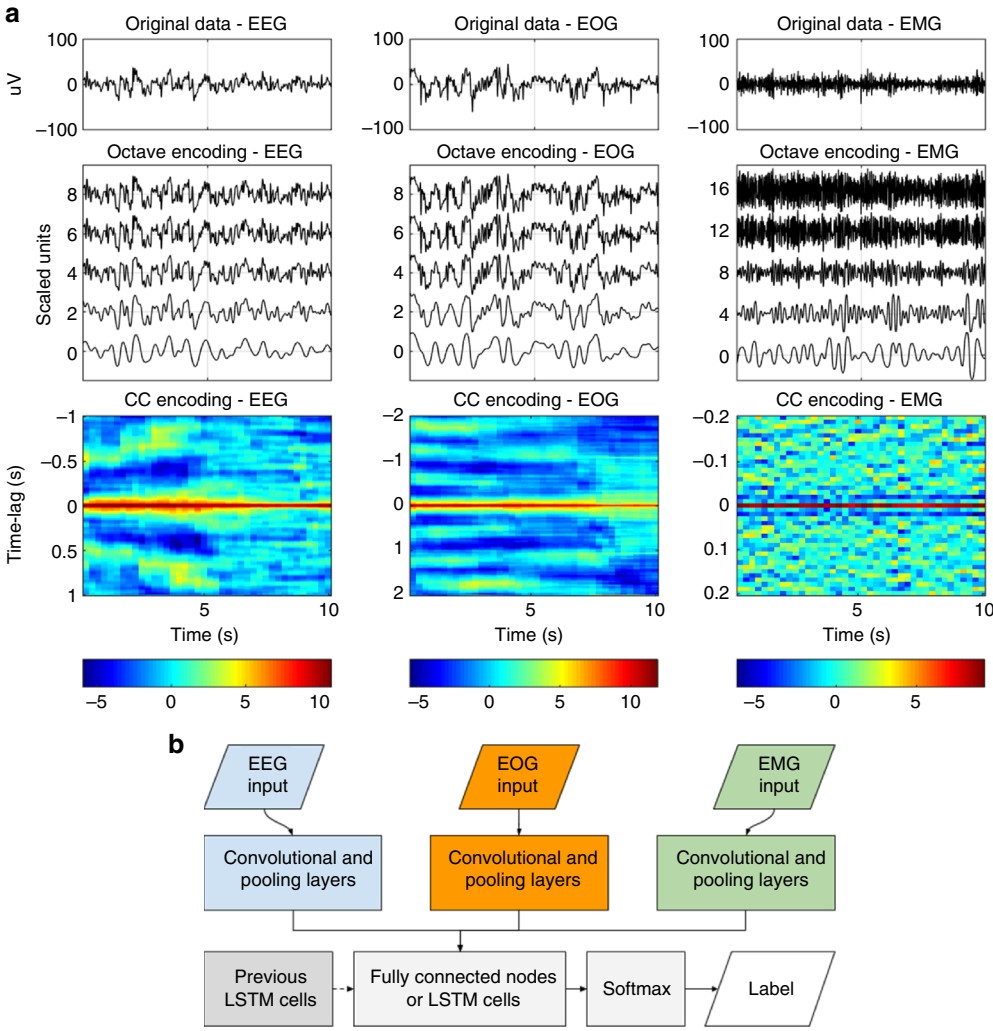

**Fig. 6** Neural network strategy. **a** An example of the octave and the CC encoding on 10 s of EEG, EOG and EMG data. These processed data are fed into the neural networks in one of the two formats. The data in the octave encoding are offset for visualization purposes. Color scale is unitless. **b** Simplified network configuration, displaying how data are fed and processed through the networks. A more detailed description can be found in Supplementary Figure 3

accuracy starts to decrease, as a sign of overfitting. For LSTM networks, dropout[68] was included, set at 0.5 while training. This ensured that model parameters generalized to the validation data and beyond. During training, data batches were selected at random. Given the stochastic nature of the training procedure, it was likely that two realizations of the same model would not lead to the same results, since models end up in different local minima. To measure the effect of this, two realizations were made of each model.

Apart from model realizations, we also investigated the effect of ensembling our sleep stage classification model. In general, ensemble models can yield higher predictive performance than any single model by attacking a classification or regression problem from multiple angles. For our specific use case, this resolves into forming a sleep stage prediction based on the predictions of all the models in the given ensemble. We tested several ensembles containing various numbers of model architectures and data encodings, as described in Supplementary Table 8.

**Performance comparisons of generated CNN models.** As stated, the influences of many different factors were analyzed. These included: using octave or CC encoding, short (5 s) or long (15 s) segment lengths, low or high complexity, with or without LSTM, and using a single or two realizations of a model. To quantify the effect of each, a $2^5$-factorial experiment was designed. This led to 32 different models (Supplementary Table 8). Comparison between models was done on a per-epoch basis.

**Hypnodensity as feature for the diagnosis of T1N.** To quantify narcolepsy-like behavior for a single recording $i$, features were generated based on a proto-feature derived from $k$-combinations of $S = \{W, REM, N1, N2, N3\}$. For the $n$th 5, 15 or 30 s segment in recording $i$, we take a single $k$-combination in the set of all

$k$-combinations, and calculate the proto-feature as the sum of the pair-wise products of the elements in the single $k$-combination, such that

$$\boldsymbol{\Phi}_n^{(i)}(\mathcal{S}_k) = \sum_{\zeta \in [\mathcal{S}_k]^2} \prod_{s \in \zeta} p\left(s|\mathbf{x}_n^{(i)}\right), \quad p \in [0, 1], \tag{9}$$

where $\boldsymbol{\Phi}_n^{(i)}$ is the proto-feature for the $n$th segment in recording $i$, $\zeta \in [\mathcal{S}_k]^2$ is a 2-tuple, or pair-wise combination, in the set of all pair-wise combinations in the $k$-combination of $\mathcal{S}$ and $s$ is a single element, or sleep stage, in $\zeta$. For $k = 1, \ldots, 5$, there exist 31 different $\mathcal{S}_k$, e.g., {Wake, REM}, {N1, N2, N3} etc., as shown in Supplementary Table 9. $p\left(s|\mathbf{x}_n^{(i)}\right)$ is the predicted probability of a 5, 15 or 30 s epoch belonging to a certain class in $\mathcal{S}$, given the data $\mathbf{x}_n^{(i)}$. For every value of $k$, 15 features based on the mean, derivative, entropy and cumulative sum were extracted, as shown in Supplementary Table 10.

**Additional features for T1N diagnosis.** In addition to above, another set of features reflecting abnormal sleep stage sequencing in T1N was investigated.

One set of such features was selected because they have been found to differentiate T1N from other subjects in prior studies[37,69–72]. These include: nocturnal sleep REM latency (REML)[10], presence of a nightly SOREMP (REML ≤15 min)[10], presence and number of SOREMPs during the night (SOREMPs defined as REM sleep occurring after at least 2.5 min of wake or stage 1) and nocturnal sleep latency (a short sleep latency is common in narcolepsy)[37]. Other features include a NREM Fragmentation index described in Christensen et al.[37]. (N2 and N3 combined to represent unambiguous NREM and N1 and wake combined to denote wake, NREM fragmentation defined as 22 or more occurrences where sustained N2/N3 (90 s) is broken by at least 1 min of N1/Wake), and the

number of W/N1 hypnogram bouts as defined by Christensen et al.[37]. (N1 and wake combined to indicate wakefulness and a long period defined as 3 min or more). In this study we also explore: the cumulative wake/N1 duration for wakefulness periods shorter than 15 min; cumulative REM duration following wake/N1 periods longer than 2.5 min; and total nightly SOREMP duration defined as the sum of REM epochs following 2.5 min W/N1 periods.

Another set of 9 features reflecting hypnodensity sleep stage distribution was also created as follows. As noted in Supplementary Figure 4, stages of sleep accumulate, forming peaks. These peaks were then used to create 9 new features based on the order of the peaks, expressing a type of transition (W to N2, W to REM, REM to N3 etc.). If the height of the $n^{th}$ peak is denoted as $\varphi_n$, the transition value $\tau$ is calculated as the geometric mean between successive peaks:

$$\tau_n = \sqrt{\varphi_n \cdot \varphi_{n+1}}. \tag{10}$$

Due to their likeness, W and N1 peaks were added to form a single type. All transitions of a certain type were added together to form a single feature. A lower limit of 10 was imposed on peaks to avoid spurious peaks. If two peaks of the same type appeared in succession the values were combined into a single peak.

**Gaussian process models for narcolepsy diagnosis**. To avoid overfitting, and at the same time produce interpretable results, a RFE algorithm was employed, as described in Guyon et al.[42]. Post screening, the most optimal features ($n = 38$) were used in a GP classifier as described above. GP classifiers are non-parametric probabilistic models that produce robust non-linear decision boundaries using kernels, and unlike many other classification tools, provide an estimate of the uncertainty. This is useful when combining estimates, but also when making a diagnosis; if an estimate is particularly uncertain, a doctor may opt for more tests to increase certainty before making a diagnosis. In a GP, a training dataset is used to optimize a set of hyper-parameters, which specify the kernel function, the basis function coefficients, here a constant, noise variance, and to form the underlying covariance and mean function from which inference about new cases are made[73]. In this case, the kernel is the squared exponential: $\sigma_f^2 \exp\left[\frac{-|\mathbf{x}-\mathbf{x}'|^2}{2l^2}\right]$. Two classes were established: narcolepsy type 1 and "other", which contains every other subject. These were labeled 1 and $-1$ respectively, placing all estimates in this range. For more information on GP in general, see the textbook by Rasmussen and Williams[73], while more information on variational inference for scalable GP classification can be found in the paper by Hensman et al.[74] and Matthews et al.[75].

**HLA-DQB1*06:02 testing**. HLA testing plays a role in T1N diagnosis, as 97% of patients are DQB1*06:02 positive when the disease is defined biochemically by low CSF hypocretin-1[5] or by the presence of cataplexy and clear MSLT findings[10]. As testing for HLA-DQB1*06:02 only requires a single blood test, models in which this feature was included were also tested. The specific feature was implemented as a binary-valued predictor, resulting in negative narcolepsy predictions for subjects with a negative HLA test result.

**High pretest probability sample**. MSLTs are typically performed in patients with daytime sleepiness that cannot be explained by OSA, insufficient/disturbed sleep or circadian disturbances. These patients have a higher pretest probability of having T1N than random clinical patients. Patients are then diagnosed with type 1 or type 2 narcolepsy, idiopathic hypersomnia or subjective sleepiness based on MSLT results, cataplexy symptoms and HLA results (if available). To test whether our detector differentiates T1N from these other cases with unexplained sleepiness, we conducted a post hoc analysis of the detector performance in these subjects extracted from both the test and replication datasets.

## Data availability

All the software is made available in GitHub at: https://github.com/stanford-stages/stanford-stages. We asked all contributing co-authors whether we could make the anonymized EDF available, together with age, sex and T1N diagnosis (Y/N). The SSC[10,32] (E.M.), the IS-RC[31] (S.T.K., C.K., P.K.S.), the KHC (S.C.H.), the HIS (G. P.), the DHS (P.J.), the FHC (Y.D.) and associated data are available at https://stanfordmedicine.app.box.com/s/r9e92ygq0erf7hn5re6j51aaggf50jly. The AHC (B. H.) and the CNC[12] (F.H.) are available from the corresponding investigator on reasonable request. The WSC[32,33] data analyzed during the current study are not publicly available due to specific language contained in informed consent documents limiting use of WSC human subjects' data to specified institutionally approved investigations. However, WSC can be made available from P.E.P. on reasonable request and with relevant institutional review board(s) approval. The JCTS[43] and AASM ISR[16] dataset are available from the corresponding institutions on reasonable request.

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

## Acknowledgements

This research was mostly supported by a grant from Jazz Pharmaceuticals to E.M. Additional funding came from: NIH grant R01HL62252 (to P.E.P.); Ministry of Science and Technology 2015CB856405 and National Foundation of Science of China 81420108002,81670087 (to F.H.); H. Lundbeck A/S, Lundbeck Foundation, Technical University of Denmark and Center for Healthy Aging, University of Copenhagen (to P.J. and H.B.D.S). Additional support was provided by the Klarman Family, Otto Mønsted, Stibo, Vera & Carl Johan Michaelsens, Knud Højgaards, Reinholdt W. Jorck and Hustrus and Augustinus Foundations (to A.N.O.). We thank Briana Gomez for helping on performing the last corrections on this manuscript.

## Author contributions

J.B.S. and A.N.O. participated in the design of the study, conducted most of the analyses and did most of the publication writing. M.O., A.A., E.B.L., H.E.M., O.C., D.P. and L.L. assisted in many of the analyses and data organization, plus helped edit the manuscript. F.H., H.Y., Y.L.S., Y.D., S.S., L.B., B.H., A.S., S.C.H., T.W.K., F.P., G.P., S.V., E.A., S.T.K., P.K.S., C.K. and P.E.P. contributed essential datasets, helped organization and gave feedback to manuscript content. P.J. and H.B.D.S. participated in the design of the study, supervised analyses and assisted publication writing. P.J. furthermore contributed datasets to the study. E.M. contributed datasets, participated in the design of the study, supervised the analyses and did most of the publication writing.

## Additional information

**Competing interests:** E.M. has received Jazz Pharmaceuticals contract, clinical trial and gift funding as principal investigator at Stanford University. He also consulted for

Idorsia and Merck, consulted or presented clinical trial results for Jazz Pharmaceuticals at congresses, this resulting in trip reimbursements and honoraria never exceeding 5000 dollars per year. G.P. has been on advisory boards for UCB, Jazz, Bioprojet and Idorsia. F.P. received a fee from UCB for speaking at a symposium, and a congress subscription from Bioprojet. The remaining authors declare no competing interests.

