## [Peer Review File · Nature Communications]

Reviewers' comments:

Reviewer #1 (Remarks to the Author):

Stephanson and colleagues have used machine learning and signal analysis methods to develop new ways of analyzing clinical sleep recordings. The overall topic is very important because the field of sleep medicine continues to use very primitive analysis techniques from the 1970's, and methods such as machine learning will likely be much more informative. This paper uses very large datasets collected in multiple labs and makes much more progress towards the goal of accurate and interpretable analysis than prior work. In fact, one could imagine that methods such as these may supplant traditional scoring by sleep lab technicians in the next decade. Though the impact on the field of sleep medicine is potentially large, I cannot evaluate the technical aspects of the machine learning.

Major concerns

- 1) In the search for biomarkers of narcolepsy, the authors mainly use combinations of previously known markers. One would hope that with machine learning, one could identify previously unappreciated but informative aspects of sleep physiology. Table 5 lists "the time taken before 5% of the sum of the product between W, N2 and REM, calculated at every epoch, has accumulated, weighed by the total amount of this sum". Apparently, this is an important feature for identifying narcolepsy, but I cannot understand what it means.
- 2) The authors report another feature indicative of narcolepsy: "presence and number of SOREMPs during the night (SOREMPs defined as REM sleep occurring after at least 2.5 minutes of wake or stage 1)". This notion of multiple nocturnal SOREMPs is novel, but the definition is unclear. In general, we think of a SOREMP (either on PSG or MSLT) as occurring soon after sleep onset. Now, the authors are defining a SOREMP as occurring after 2.5 minutes of wake or N1. Many patients wake during the night and then eventually enter REM sleep, so the specificity of this to narcolepsy is unclear to me. Also, using the term SOREMP with this new meaning will likely confuse many readers who still think of a SOREMP as a MSLT feature.
- 3) EEG was analyzed from central or occipital leads, but these cortical regions have different patterns of activity across sleep stages (e.g. occipital alpha when awake, spindles and slow waves more prominent in central leads in NREM sleep). Simply selecting the least noisy channel seems messy and hard to justify when the central and occipital patterns are so different. Also, what were the reference electrodes for C and O leads?
- 4) Line 693: It is hard to imagine machine-learning analysis of the PSG will make the MSLT unnecessary as this test provides objective evidence of daytime sleep propensity which impacts clinical management.

Minor concerns

- 1) The writing is generally clear, but some of the Methods will be tough to understand for many readers. Please provide more explanation or at least references for less familiar aspects of the methods that may be unfamiliar to general readers (e.g. perceptrons, softmax layer, bagging models, confusion matrix, Shannon entropy of the REM sleep stage distribution, etc). Similarly, phrases such as "stages of sleep accumulate, forming peaks" are hard to understand.
- 2) Line 56: This article will be read by many in Europe, so it would be helpful to present prevalence of sleep disorders in general rather than just in the US.
- 3) Line 64: Narcolepsy type 1 is not the "only sleep disorder" with a known pathophysiology. In addition, narcolepsy is hypothesized to be caused by an autoimmune process, but the true cause remains unclear. Same thing on line 664.
- 4) Line 119: PSG plus MSLT takes about 24 hours in total.
- 5) Line 573: Please state explicitly what you learned by using shorter resolutions.
- 6) The text contains several minor spelling or grammatical errors.

Reviewer #2 (Remarks to the Author):

The work automates sleep stage scoring to the level of experts in sleep recordings using neural networks.

The authors claim that their model performs better than any individual expert. I have three suggestions that would increase the strength of this claim. First, in its current form, the work does not have confidence intervals on the model performance or on expert performance and thus could benefit from significant testing. Second, the way the model is evaluated is not symmetric to the way experts are evaluated. Instead of comparing the model to 6 experts, and each expert to 6 experts (with the inflation is recognized), each label (expert or model) can be compared to the majority vote of the 7 experts. This will allow for a direct comparison and possible claims about significance. Third, the statement of ML getting to expert level performance on tasks (lines 124, 614) could use more and better references (healthcare, image, speech).

The authors report their results in terms of inter-scorer reliability results and in terms of model performance. I think the evaluation methodology could be made clearer in four ways. First, I'd recommend sticking to either 'accuracy' or 'agreement' rather than using them interchangeably: your abstract uses both of them to refer to model and expert, and it could be more obvious that they're supposed to be a head-to-head comparison. Second, I didn't understand the evaluation in the case of a tie: what does it mean to "split an epoch"? Is the model's output also split in the epoch, or can only thus receive a maximum partial score? I think this section would benefit from a clearer rewrite. Third, 'adjusted' and 'un-adjusted' are not mentioned in 497-503, but is referenced in 507. Fourth, is this way of evaluation standard? Could you cite where else such an approach has been taken? If the important claim of this paper is model > expert performance, then I think the evaluation methodology should be more symmetric and explained well for both.

The authors use machine learning to make sleep signal classifications. I have a few concerns regarding the methodology, and a few recommendations to make the use of ML more convincing to an ML audience. First, the writing style when describing machine learning could use improvement: "CNN is...not optimized to the task at hand" (280), "features produced reflecting properties of the data" (284-285), and "models that incorporate memory" (290), and "to ensure quick convergence" (259) are not completely wrong, but would not establish credibility with an ML audience: I recommend an ML expert rewrite the ML/DL sections. Second, the claim "adding memory improves models" is concerning: consider that CNNs can already mimic the memory behavior of LSTMs by using surrounding context (see ECG classification, for instance: <https://arxiv.org/abs/1707.01836>). I'm not sure I understand how the LSTM is incorporated into the prediction: is the CNN making a single value prediction, or is it outputting many outputs over the time window? I'd feel that using the CNN on a longer time window would have a similar effect as having LSTMs in the final layer, thus making the memory extension less useful. The network architecture (with the memory extension) could in general be explained more clearly. Third, I take concern with the use of terminology "hypnodensity graph", because to a machine learning audience, the output of the network is simply a multi-label probability distribution (if I have understood correctly), and the terminology made me think this was being introduced as a novel idea, which it is not. I was also curious why the multi-class approach wasn't taken — is it just so that the graph can be used as features for narcolepsy detection? Fourth, I recommend describing ensembling in a more ML-standard way: "Neural networks are essentially large ensembles of small models, consisting of individual perceptrons" sounds like a dated way of describing neural networks (I'd remove this line, and rewrite the rest of the paragraph). Fifth, from a methodology perspective, I recommend that the ensemble is done after the softmax rather than before, making the ensemble a wrapper around the model rather than necessitating tweaking within the model. Sixth, I don't understand the use of the hyperparameter sweep to produce uncertainties: why not just train the same model and have the stochastic nature of training give you the different models (or alternatively, report the mean and variance over different ensembles)? Seventh, on the GP

classification, I think a couple of lines on the feature elimination methodology employed are needed, and a motivation for the use of GP over a sparse linear model, for instance — even though GP is described, it's not clear to me that a linear model wouldn't have done just as well.

In general, I think the work, updated with the suggestions, has the potential to be an interesting application of machine learning to healthcare.

Reviewer #3 (Remarks to the Author):

Stephansen et al. introduce an automated sleep scoring method using neural networks which can be additionally applied in the diagnosis of narcolepsy. This is interesting and timely work with great potential for application. As far as illustrated, the methods seem sound. Importantly, the authors have kept data to train and test their neural networks/classifiers separate, and in the case of automated sleep scoring use a completely independent dataset to validate model performance. What is particularly convincing about the work is that the neural network models for automated sleep scoring make the same "errors" as human scorers, and overall seem to behave similarly than trained human individuals. In addition to the obvious benefits of having automated sleep scoring procedures, the possibility to easily obtain smaller segment sleep scoring following the traditional standards granted by this approach may find ample application in both basic research and clinical settings. The identification of a narcolepsy biomarker that can be easily obtained using only one night of PSG recordings and the methods developed by the authors is an unexpected plus, but the scoring algorithm by itself, equally deserves publication.

II. 82 f.: The statement that N3 is scored if more than 30% of an epoch are slow high amplitude waves is incorrect. The correct threshold is $> 20\%$. Same goes for similar statement in the discussion II. 658 f

What were the exact criteria for labeling a patient as type-1 narcolepsy vs. not, and which data entered training? In most of the studies, high risk individuals that might show similar sleep patterns were included. Did they qualify as non-type-1 narcolepsy and were they included in training and/or test? Was train test allocation determined randomly (also compare comment below)?

How exactly were data from the different datasets distributed/selected for training and test in the narcolepsy biomarker part? Is there a chance that the different occurrence rates of narcolepsy type-1 in samples recorded at different sites/scored by different scorers may have biased classification?

For the narcolepsy biomarker, some datasets seem to have selectively entered training only. There is no dataset that selectively entered test only. Why?

What is the performance of the developed narcolepsy diagnosis tool if one data set were taken out of training completely and used as an independent test set? My main question is: Is the model sufficiently trained to generalize to data recorded in a way the model "has not experienced" before? This would be important regarding application of the tool.

Which form does the outcome of HLA testing take? Did HLA testing enter as a dichotomous or continuous variable?

Table 6: In narcolepsy biomarker detection, what does HLA vs. HLA optimized refer to?

One strength of having a completely independent test set (e.g. using recordings made at a previously not included site completely different scorers...) is that it shows the model can generalize to completely new data. I wonder how the algorithm would perform on a more diverse test set, instead of the very homogenous test set used here to report the main results (IS-RC).

Though collected at three different sites, the sample was all female. Women are known to have more prominent spindle activity. How does the best model perform on sleep data recorded in male participants compared to scorers? Given the current procedure, there might be a slight chance that the selection of the best model might be in a way “overfitted” to the test set (sic), and that this decision might have played out differently, given another independent test set. Can the authors in addition report performance on a more diverse independent test set?

Was data from Stanford University that entered IS-RC also part of the SSC cohort? If not originally part of the same set, was it recorded at a similar time? Was the scoring effort by the 10 independent scorers (and data recording) pre-existing for the 70 datasets or conducted specifically for the current work on automated sleep scoring?

As mentioned before, thresholds for scoring that are stated both in introduction and discussion are incorrect. Was scoring in any of the samples conducted using this erroneous 30% threshold? Though it does not appear so, did information about % slow waves threshold enter in the classification procedure at any point?

The datasets seem to contain comparably little N3 sleep, which is expected to be about 20% of the night, not only between 5% and 10%. Is this due to the age of the patients/controls? How would the models perform in data that show a sleep stage distribution more similar to that of young healthy controls? Can such a sample be obtained or a post-hoc analysis be conducted?

Figure 4 is particularly interesting. “Worse” performance of the models on smaller segments probably relates to the fact that on a smaller scale, sleep stages might fluctuate frequently within each 30 s epochs that the gold standard scoring is obtained from. Might the neural networks be closer to the ground truth on these segments? Would the authors care to speculate how consistent/reliable the scoring of their algorithms is, on its own, in these smaller segments, and how this additional application may be useful for basic research and clinical applications?

Minor:

II. 111 f.: Please note that deep learning is not the only methods of machine learning that is used for image recognition/data analysis. Phrasing is misleading.

II. 183 ff.: Description of AHC: “As almost all subjects had two sleep recordings, two subset test replication samples were developed using independent sleep studies in the 2 sub-cohorts.” It does not become clear how exactly the data was treated and what/which results “two subset test replication samples” as well as “2 sub-cohorts” refers to.

II. 200 ff.: Description of JCTS, IHC, DHC, AASM do, contrary to others, not include statement on how data was used. The authors may want to keep this consistent.

II. 233 f sentence structure seems incorrect.

In Suppl Fig. 2, the notation does not seem to be completely consistent. E.g. octave models, cells in 3rd row of flow chart.

Table 5: Please state features most frequently selected for what/in which analysis

Table 6: what does N PSG refer to? Sum of training and testing PSGs? Only testing PSGs?

Suppl Table 4: More information on how to interpret this table might be useful since it shows important information on model performance.

We the authors would like to express our sincerest gratitude to the editor and the reviewers for their time and consideration regarding our manuscript for submission in Nature Communications. We deeply appreciate the insightful comments from the reviewers and we have taken them under consideration in the effort of improving the final paper. Following this paragraph is a point by point rebuttal to each comment by each reviewer.

Reviewers' comments:

Reviewer #1 (Remarks to the Author):

Stephansen and colleagues have used machine learning and signal analysis methods to develop new ways of analyzing clinical sleep recordings. The overall topic is very important because the field of sleep medicine continues to use very primitive analysis techniques from the 1970's, and methods such as machine learning will likely be much more informative. This paper uses very large datasets collected in multiple labs and makes much more progress towards the goal of accurate and interpretable analysis than prior work. In fact, one could imagine that methods such as these may supplant traditional scoring by sleep lab technicians in the next decade. Though the impact on the field of sleep medicine is potentially large, I cannot evaluate the technical aspects of the machine learning.

Major concerns

1) In the search for biomarkers of narcolepsy, the authors mainly use combinations of previously known markers. One would hope that with machine learning, one could identify previously unappreciated but informative aspects of sleep physiology. Table 5 lists "the time taken before 5% of the sum of the product between W, N2 and REM, calculated at every epoch, has accumulated, weighed by the total amount of this

sum”. Apparently, this is an important feature for identifying narcolepsy, but I cannot understand what it means.

As the reviewer mentioned, some of the features identified are non-trivial and novel. This is the interest of machine learning. In this particular instance, every epoch gives a probability for each sleep stage. The sum of the product at each epoch is

$$p(W) \times p(N2) + p(W) \times p(REM) + p(N2) \times p(REM).$$

This sum is calculated for all epochs of the study and accumulated, reaching a value X. The time it takes between sleep onset and this product reaching 5% of the total sum X over the night is the value that is used for the feature. Clinically, this can be interpreted as an uncertainty or transition feature between wakefulness, REM and N2 sleep, since each addend in the summation reflects the uncertainty/transition between each sleep stage pair. Eg: for sleep stage *i* the model predicts W and N2 with equally high probability with all other stages being low probability. The epoch sum product will then reflect this uncertainty by having the maximum value if and only if the two stages are equally probable. If the probability for W decreases, while the probability for N2 increases, the sum-product value above will decrease in value all other stage probabilities being low. A study with a high dissociation between sleep and wake at the beginning of sleep will correspondingly be reflected in a smaller feature value.

However, we also agree with the reviewer comment, that markers that have no obvious meaning could be confusing to clinicians. For this reason, we are now creating a narcolepsy predictor web site where not only the result of the machine learning predictor will be displayed, but also result of more traditional markers such as REML during the PSG and SOREMPs during the night, following past publication data by Christensen et al. (2015)¹.

2) The authors report another feature indicative of narcolepsy: “presence and number of SOREMPs during the night (SOREMPs defined as REM sleep occurring after at least 2.5 minutes of wake or stage 1)”. This notion of multiple nocturnal SOREMPs is novel, but the definition is unclear. In general, we think of a SOREMP (either on PSG or MSLT) as occurring soon after sleep onset. Now, the authors are defining a SOREMP as occurring after 2.5 minutes of wake or N1. Many patients wake during the night and then eventually enter REM sleep, so the specificity of this to narcolepsy is unclear to me. Also, using the term SOREMP with this new meaning will likely confuse many readers who still think of a SOREMP as a MSLT feature.

This is a good point and two separate aspects should be briefly discussed. First, with regard to using the term “number of SOREMP during the night”, the feature has been described in a prior publication and by others^{1,2} so we believe it should not be confusing to the reader. In these publications, it was found that narcolepsy patients would wake up or go to stage 1 for periods of time and then go straight to REM sleep without going through stage 2. This feature has been found by others, who also called it “SOREMPs” at night. Second, in this particular case we are treating stage 1 as it was wake.

¹ Christensen, J. A. E. et al. Sleep-stage transitions during polysomnographic recordings as diagnostic features of type 1 narcolepsy. *Sleep Med.* 16, 1558–1566 (2015).

² 1. Drakatos, P. et al. First rapid eye movement sleep periods and sleep-onset rapid eye movement periods in sleep-stage sequencing of hypersomnias. *Sleep Med.* 14, 897–901 (2013).

As this reviewer perhaps knows, there is quite a bit of controversy on what stage 1 exactly represent. It is an intermediary stage (with low inter-rater reliability) where some mental activity occurs, a phenomenon which has led some authors to consider that sleep onset with loss of consciousness only occurs during stage 2. This led us to add this feature and other related features in the Gaussian predictor study. The reasoning behind this feature is detailed in the section ‘Additional Features for Type 1 Narcolepsy diagnosis’ in the Methods section.

3) EEG was analyzed from central or occipital leads, but these cortical regions have different patterns of activity across sleep stages (e.g. occipital alpha when awake, spindles and slow waves more prominent in central leads in NREM sleep). Simple selecting the least noisy channel seems messy and hard to justify when the central and occipital patterns are so different. Also, what were the reference electrodes for C and O leads?

We apologize if this was not clear. Both central and occipital leads were referenced to the contralateral mastoid or auricular process. However, it is important to note that the noise selection process is not performed to choose between a central and occipital derivation, but rather to choose between left/right hemisphere derivations.

4) Line 693: It is hard to imagine machine-learning analysis of the PSG will make the MSLT unnecessary as this test provides objective evidence of daytime sleep propensity which impacts clinical management.

We are certainly only talking about the MSLT in the context of narcolepsy type 1. If a patient has clear cataplexy, it is reasonable to assume that maybe the MSLT is not needed. It is an entirely different story in cases without cataplexy. This caveat has now been added in the discussion.

Minor concerns

1) The writing is generally clear, but some of the Methods will be tough to understand for many readers. Please provide more explanation or at least references for less familiar aspects of the methods that may be unfamiliar to general readers (e.g. perceptrons, softmax layer, bagging models, confusion matrix, Shannon entropy of the REM sleep stage distribution, etc). Similarly, phrases such as “stages of sleep accumulate, forming peaks” are hard to understand.

We thank the reviewer for bringing this to our attention. Of course, we would like to accommodate this request, but then it becomes a question of how to decide on generally accepted knowledge. Some concepts, while very foreign for some readers, will likely be trivial to others, making it impossible to cater to all. We feel we have introduced sufficient explanations for more complicated concepts, while providing enough references for further reading. For general deep learning concepts, we have referred readers to the review article published in Nature by LeCun, Bengio and Hinton (2015) and the deep learning textbook by Goodfellow, Bengio and Courville (2016), while for general machine learning concepts we have referred readers to the excellent textbook by Bishop (2006).

2) Line 56: This article will be read by many in Europe, so it would be helpful to present prevalence of sleep disorders in general rather than just in the US.

We added references to review papers on narcolepsy that includes prevalence numbers pertaining to Europe and the Asia.

3) Line 64: *Narcolepsy type 1 is not the “only sleep disorder” with a known pathophysiology. In addition, narcolepsy is hypothesized to be caused by an autoimmune process, but the true cause remains unclear. Same thing on line 664.*

These paragraphs have now been rephrased accordingly.

4) Line 119: *PSG plus MSLT takes about 24 hours in total.*

This was changed in the text.

5) Line 573: *Please state explicitly what you learned by using shorter resolutions.*

In the section **Model performance** under **Results**, we state that varying the segment length (ie. 5s or 15 s resolutions) were of less importance than encoding (correlation vs. octave) and memory (incorporating LSTM layers or not), which is also shown in Figure 3A. Prominent interaction effects are then detailed and shown in Supplementary Figure 3, where it can be seen that longer resolutions show slightly higher performance on average than shorter resolutions, but that the effect is more prominent in lower complexity models. Furthermore, the model performances in terms of accuracy versus the scoring resolution is also shown in Figure 4b.

6) *The text contains several minor spelling or grammatical errors.*

We have asked a native speaker to reread and correct all possible errors.

Reviewer #2 (Remarks to the Author):

The work automates sleep stage scoring to the level of experts in sleep recordings using neural networks.

The authors claim that their model performs better than any individual expert. I have three suggestions that would increase the strength of this claim. First, in its current form, the work does not have confidence intervals on the model performance or on expert performance and thus could benefit from significant testing.

We have now changed the way we perform this analysis. For each individual scorer, we create a consensus scoring of the remaining 5 scorers and test the performance of this individual compared to the ensemble model performance on the same consensus using a paired t-test. This is performed for each individual scorer. Our results show that the model is performing significantly better on unbiased consensus scorings for all individual scorers.

Second, the way the model is evaluated is not symmetric to the way experts are evaluated. Instead of comparing the model to 6 experts, and each expert to 6 experts (with the inflation is recognized), each label (expert or model) can be compared to the majority vote of the 7 experts. This will allow for a direct comparison and possible claims about significance.

We are sorry for the misunderstanding. In fact, we are doing exactly that. The total number of expert is six only, and each label (expert or model) is compared to the majority vote of the 6 experts in row 1 of Table 1. However, to avoid the fact the consensus may contain data from a scorer also used in the consensus, we also compared each scorer to the consensus of the other 5 scorers (second row of table 1). As this was not clearly mentioned, we changed the legend in Table 1 accordingly. On addition, we also studied the majority vote of 2, 3, 4 and 5 and 6 experts to that of the machine learning prediction which is independent of any scorer data (figure 4A).

Third, I the statement of ML getting to expert level performance on tasks (lines 124, 614) could use more and better references (healthcare, image, speech).

We have now added 6 more references on machine/deep learning applications to radiology, pathology and retinopathy.

The authors report their results in terms of inter-scorer reliability results and in terms of model performance. I think the evaluation methodology could be made clearer in four ways.

First, I'd recommend sticking to either 'accuracy' or 'agreement' rather than using them interchangeably: your abstract uses both of them to refer to model and expert, and it could be more obvious that they're supposed to be a head-to-head comparison.

The distinction between 'accuracy' and 'agreement' is subtle, but intentional. 'Agreement' refers specifically to the consensus labeling of the multiple scorers, ie. the 'gold standard', while 'accuracy' is a performance metric that relates the model or individual scorer performance to the consensus.

Second, I didn't understand the evaluation in the case of a tie: what does It mean to "split an epoch"? Is the model's output also split in the epoch, or can only thus receive a maximum partial score? I think this section would benefit from a clearer rewrite.

The reviewer is absolutely right, the wording of the paragraph is a bit confusing for some readers. The specific question about splitting an epoch simply refers to the fact that in case of a tie, the epoch is counted equally those classes contributing to the tie.

Third, 'adjusted' and 'un-adjusted' are not mentioned in 497-503, but is referenced in 507. Fourth, is this way of evaluation standard? Could you cite where else such an approach has been taken? If the important claim of this paper is model > expert performance, then I think the evaluation methodology should be more symmetric and explained well for both.

In the un-adjusted consensus, each scorer contributes equally to the estimated gold standard. However, because we have six scorers, this may result in a tie between two or three stages. In the event of this, the estimated value is divided between each stage involved in the tie, as well as their corresponding incorrect estimates. Example: Three scorers estimate N3 and three scorers estimate N2. In this situation the epoch estimate is divided as 50% N2 and 50% N3. In the confusion matrix, this will correspond to 25% true N2, 25% true N3, 25% N2 incorrectly estimated as N3 and vice versa. The adjusted consensus avoids this by placing increased weight on scorers that on average perform better, determined by the scorers Cohen's kappa. Example: Three scorers estimate N3 and three scorers estimate N2. However, because the scorers who estimate N3 on average perform

better, the is estimated as 100% N3. In the confusion matrix, this will correspond to 50% true N3, and 50% N3 incorrectly estimated as N2.

The authors use machine learning to make sleep signal classifications. I have a few concerns regarding the methodology, and a few recommendations to make the use of ML more convincing to an ML audience.

First, the writing style when describing machine learning could use improvement: “CNN is...not optimized to the task at hand” (280), “features produced reflecting properties of the data” (284-285), and “models that incorporate memory” (290), and “to ensure quick convergence” (259) are not completely wrong, but would not establish credibility with an ML audience: I recommend an ML expert rewrite the ML/DL sections.

We have asked a resident ML expert to look over the methods section.

Second, the claim “adding memory improves models” is concerning: consider that CNNs can already mimic the memory behavior of LSTMs by using surrounding context (see ECG classification, for instance: <https://arxiv.org/abs/1707.01836>). I’m not sure I understand how the LSTM is incorporated into the prediction: is the CNN making a single value prediction, or is it outputting many outputs over the time window? I’d feel that using the CNN on a longer time window would have a similar effect as having LSTMs in the final layer, thus making the memory extension less useful. The network architecture (with the memory extension) could in general be explained more clearly.

Regarding the specific paper the reviewer cited from the Stanford ML group on automatic classification of ECG rhythms: what the authors are proposing in their work is actually very similar to what we are proposing. In fact, it is equivalent in the cases where we have tested feed-forward (FF) layers in the final prediction layer, albeit with a variant ResNet-34 CNN structure described by He et al. (2016), while we are using a variant of the VGG network first presented by Simonyan and Zisserman (2015). The network presented in the paper cited by the reviewer is predicting a label each second, but the reviewer is incorrect in stating that the CNN architecture is mimicking LSTM networks, since the 1 s predictions are independent of previous predictions and thus have no knowledge of long-term dependencies. Indeed, our network essentially consists of two parts: a data-driven feature extractor (the CNN network) and a prediction layer (either FF or LSTM). For each time step (5, 10 or 15 s windows) in a five-minute sequence of training data, the CNN network essentially outputs a feature vector, which is then fed to the LSTM network. We present results both in the main text and supplementary material that are indeed showing superior performance of adding memory, ie. LSTM cells, to our model. Research from certain DL groups have shown results that indicate that CNNs can eliminate LSTMs in some use cases, however, this is the exception rather than standard accepted practice. Although we agree that the description in the text body of the network architecture could be improved, we also refer to the supplementary figure 2 for further details of the network, which includes input and output dimensions for each layer along with the general flow of data through the network.

Third, I take concern with the use of terminology “hypnodensity graph”, because to a machine learning audience, the output of the network is simply a multi-label probability distribution (if I have understood correctly), and the terminology made me think this was being introduced as a novel idea, which it is not.

We entirely agree reporting on a multi-label probability distribution is not a new concept. However, for sleep researchers and clinicians, this is an entirely new presentation of the data versus conventional hypnograms that

has real clinical value. We could use “multi-label sleep stage probability distribution hypnogram” but this would be very cumbersome. For this reason, we are keeping this term intact, although also mentioning multi-label sleep stage probability distribution as the correct way to state what is plotted, and the term hypnodensity as a simpler denotation.

I was also curious why the multi-class approach wasn't taken — is it just so that the graph can be used as features for narcolepsy detection?

The reviewer raises an excellent point, thank you. The reason for this approach is partly based on the way the project evolved over time. There is significant clinical and research value in the hypnodensity graph alone, towards which is what we originally were working. As the project developed, we realized we could use this for another purpose, ie. narcolepsy prediction, however, we do not make the claim that the current iteration of the narcolepsy detection algorithm using the Gaussian process model is the best nor most efficient. Essentially, the narcolepsy prediction could (and probably should!) also be made based on the hypnodensity graph using a separate layer of recurrent neural networks in either a long short-term memory or gated recurrent unit configuration. We found that the Gaussian process classification algorithm was working well for our problem and also contains nice properties such as variance estimation on predictions. We also applied lasso logistic regression and XGBoost, a recently developed machine learning algorithm very popular in the Kaggle community, to our problem, but we did not obtain as good results as with the Gaussian process model.

Fourth, I recommend describing ensembling in a more ML-standard way: “Neural networks are essentially large ensembles of small models, consisting of individual perceptrons” sounds like a dated way of describing neural networks (I'd remove this line, and rewrite the rest of the paragraph).

We agree that the paragraph might be confusing to some readers, as the purpose of the paragraph was to justify testing ensembles consisting of different model configurations and not so much about neural networks in general. We have rewritten the paragraph to focus more on ensembling our models.

Fifth, from a methodology perspective, I recommend that the ensemble is done after the softmax rather than before, making the ensemble a wrapper around the model rather than necessitating tweaking within the model.

We apologize for any confusion regarding the methodology; the ensembling operation is indeed performed after the softmax independently of each model. To clarify, each model takes as input 2 EEG channels, 2 EOG channels and 1 EMG chin channel. These are subjected to data processing algorithms, ie. the octave or correlation decompositions for each channel and for each time epoch (5, 10 or 15 s) in a five-minute segment, and the resulting data representations are then fed to parallel processing using the CNN structure. The three outputs from the CNN (one output vector for each modality, ie. EEG, EOG and EMG) are then collected and fed to the memory layers, which finally results in a prediction for each time step. The outputs for all 16 models are then collected in the final ensemble. We agree, that in the original text, the paragraph is confusing, and we have not rewritten the paragraph to better convey the ensembling structure.

Sixth, I don't understand the use of the hyperparameter sweep to produce uncertainties: why not just train the same model and have the stochastic nature of training give you the different models (or alternatively, report the mean and variance over different ensembles)?

The reviewer makes an excellent point in this regard; it is an entirely feasible and reasonable approach to train the same model and rely on the training to differentiate the models. However, to ensure heterogeneity across the models, we chose to implement the parameter sweeps instead. This way, we can maximize the differences between the models within the confines of the model structure itself, and therefore maximize the effectiveness of the final 16-model ensemble. The (currently unverified) hypothesis is that by doing so, the specific models will focus on different aspects in the signal data that results in the same conclusion, ie. the correct sleep stage. This can be researched further using sensitivity analysis, activation maximization or deep Taylor decomposition algorithms, which is also an active area of research³, although validation of this for the current project exceeds the scope of this paper. However, we are actually also investigating the effect of multiple realizations as mentioned in paragraph 2 of the section ‘Training of CNN models’.

Seventh, on the GP classification, I think a couple of lines on the feature elimination methodology employed are needed, and a motivation for the use of GP over a sparse linear model, for instance — even though GP is described, it’s not clear to me that a linear model wouldn’t have done just as well.

The reviewer raises a fair point. A Gaussian process defines a distribution over functions, which can be used for regression or classification. This distribution is parameterized by a mean function and a covariance function aka. a kernel. Depending on the application there are many different options. A common choice of kernel is the squared exponential with an added Gaussian noise kernel which has some advantageous properties:

- a) The squared exponential kernel has good smooth and non-linear properties, which allows it to fit most non-linear functions. In addition, by using a unique length scale hyperparameter for each feature in the kernel, we can automatically determine the relevance of each feature. The kernel kernel is very intuitive, in that observations which are near each other in input space are highly correlated, while observations that are far from each other are not.
- b) The Gaussian noise kernel allows us to explicitly model the underlying noise or un-accounted-for variance in the data, which can then be directly interpreted.

These properties are desirable for further analysis, but the reviewer is correct in stating that a simple linear model could also be used. Referring to a previous question by the same reviewer, we did investigate other classification algorithms, namely the XGBoost algorithm and L1-regularized logistic regression, but we did not obtain as good results using those methods. Our interpretation of this is that the ability to introduce uncertainties in the classification algorithm is a powerful property that lends itself well to data of biological origin.

For more information on the underlying algorithm, we refer to Hensman (2015)⁴ and the paper behind the GPflow library for Gaussian process classification⁵. We have added these references to the paper for those readers that are interested.

In general, I think the work, updated with the suggestions, has the potential to be an interesting application of machine learning to healthcare.

We thank the reviewer for the excellent review comments.

³ See eg. A. Vilamala, K. H. Madsen, L. K. Hansen: Deep Convolutional Neural Networks for Interpretable Analysis of EEG Sleep Stage Scoring. In IEEE 2017 International Workshop on Machine Learning for Signal Processing, September 25-28, 2017, Tokyo, Japan for an application of sensitivity analysis for interpretation of spectrograms of sleep stages.

⁴ J. Hensman, A. G. de G. Matthews, Z. Ghahramani: Scalable Variational Gaussian Process Classification. In Proceedings of the 18th International Conference on Artificial Intelligence and Statistics (AISTATS) 2015, San Diego, CA, USA.

⁵ A. G. de G. Matthews, M. van der Wilk, T. Nickson, K. Fujii, A. Boukouvalas, P. León-Villagrà, Z. Ghahramani, J. Hensman: GPflow: A Gaussian Process Library using TensorFlow. Journal of Machine Learning Research, 40, pp. 1-6, 2017.

Reviewer #3 (Remarks to the Author):

Stephansen et al. introduce an automated sleep scoring method using neural networks which can be additionally applied in the diagnosis of narcolepsy. This is interesting and timely work with great potential for application. As far as illustrated, the methods seem sound. Importantly, the authors have kept data to train and test their neural networks/classifiers separate, and in the case of automated sleep scoring use a completely independent dataset to validate model performance. What is particularly convincing about the work is that the neural network models for automated sleep scoring make the same “errors” as human scorers, and overall seem to behave similarly than trained human individuals. In addition to the obvious benefits of having automated sleep scoring procedures, the possibility to easily obtain smaller segment sleep scoring following the traditional standards granted by this approach may find ample application in both basic research and clinical settings. The identification of a narcolepsy biomarker that can be easily obtained using only one night of PSG recordings and the methods developed by the authors is an unexpected plus, but the scoring algorithm by itself, equally deserves publication.

ll. 82 f.: The statement that N3 is scored if more than 30% of an epoch are slow high amplitude waves is incorrect. The correct threshold is > 20%. Same goes for similar statement in the discussion ll. 658 f

We are sorry for this mistake. It is indeed 20% and has been corrected in both the introduction and the discussion.

What were the exact criteria for labeling a patient as type-1 narcolepsy vs. not, and which data entered training? In most of the studies, high-risk individuals that might show similar sleep patterns were included. Did they qualify as non-type-1 narcolepsy and were they included in training and/or test? Was train test allocation determined randomly (also compare comment below)?

The detailed descriptions on how patients in these cohorts are included and diagnosis criteria are described in Supplementary material, cohort description and supplementary table 1. Type 1 narcolepsy cases were all either with clear cataplexy and positive MSLTs or with low CSF hypocretin-1 (strict gold standard). In the training set, hypersomnia cases were without cataplexy with or without a positive MSLT or with normal CSF hypocretin. In the confirmation set, a few cases with Type 2 narcolepsy are included (no cataplexy and positive MSLT) but are still labeled as “other hypersomnia.” We are sorry that this was not clear and this has been precised. The low frequency of type 2 narcolepsy in training dataset, plus the fact only a small portion of these could be narcolepsy type 2 with hypocretin deficiency, explain the high performance.

How exactly were data from the different datasets distributed/selected for training and test in the narcolepsy biomarker part? Is there a chance that the different occurrence rates of narcolepsy type-1 in samples recorded at different sites/scored by different scorers may have biased classification?

This varies a lot and is described in details in Supplementary material, cohort description and supplementary table 1. For obvious reasons, some cohort have been intentionally enriched in atypical, high pretest probability narcolepsy cases (other hypersomnia and narcolepsy without cataplexy) as it is in this context that the diagnosis tool will be the most helpful. It is for this reason that we are also presenting Figures 1E and 1F, which look at

performance in narcolepsy versus other hypersomnias. Because in this case the number of “other hypersomnia” cases is limited, and considering the fact there is minimal overfitting in training versus test and replication sets (Figure 1 A versus 1B and 1C), we pulled Type 1 and other hypersomnia cases from both the test and the replication set for this analysis.

For the narcolepsy biomarker, some datasets seem to have selectively entered training only. There is no dataset that selectively entered test only. Why?

It is due to logic and how the project developed, and new datasets were added. It must be considered that to retrain these models on everything takes a huge amount of computer power and time, and thus in many cases it was just deemed unnecessary.

For sleep scoring training and testing, we did not want to include too many narcoleptic patients, but only include some to the extent that they will be represented in sleep clinics. This is why the sleep stage training is mostly using the Wisconsin sleep cohort and the Stanford Sleep Cohort (any sleep patient, mostly sleep apnea, a handful of narcolepsy cases) that were used for training. For sleep stage testing, we used samples from the same cohorts plus added the Korean sleep sample that contains more narcoleptic and hypersomnia subjects to include an independent dataset, and because we were starting to get interested in a narcolepsy application. Finally, it is fairly obvious why we “reserved” the multiscore dataset (IS-RC) only for testing the sleep scoring model. To include this data in training would have been a waste considering how useful it is to check performance of machine learning versus interindividual scorer.

For developing the narcolepsy biomarker, we purposefully enriched the datasets with patients with type 1 narcolepsy, and hypersomnia, since this is where the use of the biomarker is the most helpful. We tried to use as many sources as possible and to split them between training and testing, adding also the controls of the Wisconsin and Stanford Sleep Cohort. One exception is the Jazz clinical trial dataset that was so small (n=7) that we decided to include it only for training and not testing, and the Danish Hypersomnia samples, that we only included for training because it was not available to us at Stanford for IRB reasons and thus was treated a bit differently because the PI of the study (E. Mignot) was unable to give directions. It should be noted that the performance is really similar in the training and testing, which shows that we have minimal overfitting of the data.

It should be noted that the detector has now also been tested on a “naïve” set of PSGs from China and France, coming from sites that had never been seen before by the detector. This greatly increase the value of the detector but also illustrate that design of studies such as these have to evolve as the project progress. We believe it is important to report findings as they are made, avoiding any bias.

What is the performance of the developed narcolepsy diagnosis tool if one data set were taken out of training completely and used as an independent test set? My main question is: Is the model sufficiently trained to generalize to data recorded in a way the model “has not experienced” before? This would be important regarding application of the tool.

This is an excellent comment that complements the answer above and is also addressed in the answer to the editor. It is accepted wisdom in the field to split all the datasets between training and testing, but the reviewer is correct in pointing out that it limits generalization to entirely new datasets. We believed that because we had

included extremely diverse datasets (from more than 10 different sleep labs), generalization to other datasets that have not been “seen” in training was likely, but this was not formally tested. In this revision, we added a new dataset containing Chinese and French controls and type 1 narcolepsy patients, showing generalization and excellent performance.

Which form does the outcome of HLA testing take? Did HLA testing enter as a dichotomous or continuous variable?

It is dichotomous, positive or negative.

Table 6: In narcolepsy biomarker detection, what does HLA vs. HLA optimized refer to?

Once HLA is added, it increases tremendously specificity, so a slightly different threshold on the ROC curves gives a better balance of sensitivity and specificity.

One strength of having a completely independent test set (e.g. using recordings made at a previously not included site completely different scorers...) is that it shows the model can generalize to completely new data. I wonder how the algorithm would perform on a more diverse test set, instead of the very homogenous test set used here to report the main results (IS-RC). Though collected at three different sites, the sample was all female. Women are known to have more prominent spindle activity. How does the best model perform on sleep data recorded in male participants compared to scorers? Given the current procedure, there might be a slight chance that the selection of the best model might be in a way “overfitted” to the test set (sic), and that this decision might have played out differently, given another independent test set. Can the authors in addition report performance on a more diverse independent test set?

As answered to the editor, in our opinion, there is no reason that having a multiscorer cohort that include both sexes (versus female only) would change the conclusion with respect to the performance of the machine learning. We used this cohort because it was carefully designed, top technicians of known and established centers were used, and results were previously published. It is noteworthy that in this and subsequent publications (Kuna et al., Sleep, 2015), the fact the cohort is only composed of middle age women is not even discussed at all as a limitation. I am not sure why only females were included, but as can be noted in supplementary table 8 there no difference in performance across sex within each pathology, thus it makes no sense to hypothesize a difference in an even smaller multiscorer sample. We have added a caveat sentence regarding the female only status of the multi scorer cohort. In general, we believe that Table 8 answer this question plainly, performance of sleep scoring varies greatly per site/cohort, and it is likely due to the quality of the scorers, although we cannot exclude the fact the machine learning also works better based on hardware settings. This was already discussed in detail in the discussion section.

If the reviewer and the editor insist, we could ask 6 new technicians to score an additional sample of men only PSGs, but we don't think it is a good idea as results would be difficult to interpret. Indeed, is a very significant difference in accuracy across cohorts-- this is again most likely due to the quality of the sleep scoring technicians at various sites (with the Korean cohort performing worse, data not shown). As a consequence, to be sure this is not a factor, we would need to use the exact same scorer, which would be impossible.

Was data from Stanford University that entered IS-RC also part of the SSC cohort? If not originally part of the same set, was it recorded at a similar time? Was the scoring effort by the 10 independent scorers (and data recording) pre-existing for the 70 datasets or conducted specifically for the current work on automated sleep scoring?

This was detailed in the supplementary material, cohort description, Patient-based Inter-scorer Reliability Cohort (IS-RC) section. What was not mentioned is that the main purpose was to compare a rule based automatic scoring algorithm for the scoring of sleep apnea. This has now been added. It should be noted that the cost of doing such a study is high, thus it would never have been feasible to generate this dataset for this study alone in the current NIH funding environment.

As mentioned before, thresholds for scoring that are stated both in introduction and discussion are incorrect. Was scoring in any of the samples conducted using this erroneous 30% threshold? Though it does not appear so, did information about % slow waves threshold enter in the classification procedure at any point?

We are very sorry for this, as mentioned above 20% was used. It does not change anything to any calculation, as it is only used to define stage 3 by the scorers.

The datasets seem to contain comparably little N3 sleep, which is expected to be about 20% of the night, not only between 5% and 10%. Is this due to the age of the patients/controls? How would the models perform in data that show a sleep stage distribution more similar to that of young healthy controls? Can such a sample be obtained or a post-hoc analysis be conducted?

Yes, the main reason for low N3 is the fact these subjects are generally older. We explore the effect of age on scoring accuracy by sleep disorder in supplementary table 8. As can be seen, age has a modest detrimental effect in most condition and overall on scoring accuracy. As discussed in the text regarding condition effects, it is impossible to say whether it is because the machine learning is performing more poorly, or whether it is because the scorers perform less well (more likely, sleep being more fragmented, scorers may be more lazy; the same may be true with narcolepsy where performance is extremely affected, most likely because it is difficult to score by humans, see discussion).

Figure 4 is particularly interesting. “Worse” performance of the models on smaller segments probably relates to the fact that on a smaller scale, sleep stages might fluctuate frequently within each 30 s epochs that the gold standard scoring is obtained from. Might the neural networks be closer to the ground truth on these segments? Would the authors care to speculate how consistent/reliable the scoring of their algorithms is, on its own, in these smaller segments, and how this additional application may be useful for basic research and clinical applications?

This is an excellent comment, and we are exactly working in this direction! Our hypothesis is that some of these smaller time scale changes could be very meaningful clinically. For example, they could be used to better define micro-arousals and their effect on daytime alertness or perception of sleep continuity. We now have developed a very nice algorithm scoring micro-arousals based on gold standard human scoring of these events, and are hoping to test if some “micro-wake epoch” or changes of sleep stages from deeper to less deep sleep stages as detected by the sleep scoring algorithm could also be used as a measure of microarousal. These could also be used to look at CAPs. These are only a few of the many applications of this research and we did not want to

speculate too much. Rather, we are providing the code and hope that more people will join in this research as there is so much to do.

Our team is dedicated to continue working in this area and to provide all the materials and code so that more people join, use and improve this work. We are working towards making available some of the PSG EDFs, although this is complicated as many are not ours to give away. Most importantly, we are just initiating a new project called the STAGES study that aims at collecting and making available 30,000 PSG EDF, with detailed phenotypic and genetic characterization. The sleep scoring codes we will develop will be used on these, but importantly will also be made available as they are being developed. We are, we believe, entering a completely new area of sleep research using PSGs and feel confident many people will join, providing good datasets and clear code are made available.

Minor:

ll. 111 f.: Please note that deep learning is not the only methods of machine learning that is used for image recognition/data analysis. Phrasing is misleading.

This was changed.

ll. 183 ff.: Description of AHC: “As almost all subjects had two sleep recordings, two subset test replication samples were developed using independent sleep studies in the 2 sub-cohorts.” It does not become clear how exactly the data was treated and what/which results “two subset test replication samples” as well as “2 sub-cohorts” refers to.

This was changed.

ll. 200 ff.: Description of JCTS, IHC, DHC, AASM do, contrary to others, not include statement on how data was used. The authors may want to keep this consistent.

This was changed. Further details on the demographics and use for each cohort is supplied in Supplementary Table 1.

ll. 233 f sentence structure seems incorrect.

This was changed.

In Suppl Fig. 2, the notation does not seem to be completely consistent. E.g. octave models, cells in 3rd row of flow chart.

The reason for the discrepancy is due to the fact that the octave encoding does not result in ‘image’ representation of the input data, as in the cross-correlation encoding. Brackets in the left side of the flowchart describe a two-dimensional kernel, while the octave models only contain a single dimension (ie. time).

Table 5: Please state features most frequently selected for what/in which analysis

The features listed in this table are more detailed descriptions of the 8 most frequently used features out of the total 38 features used in the Gaussian process model.

Table 6: what does N PSG refer to? Sum of training and testing PSGs? Only testing PSGs?

This refers to the total number of PSGs underlying the performance metrics. This has been changed in the table. All models are based on the training data and then tested on the samples listed in the first column.

Suppl Table 4: More information on how to interpret this table might be useful since it shows important information on model performance.

This was extended.

REVIEWERS' COMMENTS:

Reviewer #1 (Remarks to the Author):

Stephansen and colleagues have investigated the use of neural networks in scoring PSG recordings, and the usefulness of these analyses in diagnosing narcolepsy type 1 (NT1). They have addressed most comments thoroughly, and the paper is now clearer. I have only a few minor concerns.

Minor concerns:

Pg 3, line 79: Narcolepsy is not a “lifelong” condition; it usually begins in childhood.

Pg 3, line 80: Prevalences are not the same globally (narcolepsy seems rare in the Middle East)

Pg 8, line 302: please state that C and O electrodes are referenced to contralateral mastoids.

Pg 8, Line 304: “EEG hemisphere” is unclear; please just say right or left side.

Pg 16, line 630: Analysis in 5 sec epochs was almost as good as 30 sec, but this does not get much discussion. I think this is an important point and could be expanded upon.

Pg 19, line 741: Some clinicians are already confused by the meaning of sleep-onset REM sleep periods (SOREMP), and the author’s use of the term SOREMP will only add to the confusion. I suggest something like “short latency REM sleep”. Also, this should be defined in the main text, not the Supplement.

Pg 20, line 770: The authors state that “The use of this novel biomarker will reduce time spent to a standard 8-hour night recording...”, but then this is somewhat contradicted by the last sentence of the paragraph which correctly points out that the MSLT also provides useful information on the propensity for a patient to fall asleep during the day. For now, the MSLT will remain important for diagnosing hypersomnias other than NT1, and I hope this sort of ML PSG analysis does not displace the MSLT in diagnosing NT1.

The text still contains a few typos:

Pg 16, line 628: should be “extent”

Pg 19, line 737: should say “...type 1 narcolepsy were...”

Pg 19, line 741: should be “...bears great resemblance...”

Pg 19, line 761: might be clearer as “A PSG with high staging uncertainty...”

Reviewer #2 (Remarks to the Author):

The authors have addressed all concerns in revised manuscript.

Reviewer #3 (Remarks to the Author):

I want to thank the authors for their responses to my previous comments. Since the exact changes made to the manuscript are not detailed in the rebuttal, I apologize if I have overlooked something that has already been done.

I agree that it will not be necessary to add a multiscore cohort that includes both sexes (referring also to Table 8), yet appreciate the note of caution added to the discussion.

The added cohorts that replicate the high accuracy on sleep stage and narcolepsy classification further strengthen this work. The authors might consider stressing that this additional replication (also in the healthy controls) further validates their detector also in a non-clinical setting.

Apart from this, I have no more concerns.

REVIEWERS' COMMENTS:

Reviewer #1 (Remarks to the Author):

Stephansen and colleagues have investigated the use of neural networks in scoring PSG recordings, and the usefulness of these analyses in diagnosing narcolepsy type 1 (NT1). They have addressed most comments thoroughly, and the paper is now clearer. I have only a few minor concerns.

Minor concerns:

Pg 3, line 79: Narcolepsy is not a “lifelong” condition; it usually begins in childhood.

This was corrected.

Pg 3, line 80: Prevalences are not the same globally (narcolepsy seems rare in the Middle East)

This was changed into “Narcolepsy affects approximately 0.03% of the US, European, Korean and Chinese population”

Pg 8, line 302: please state that C and O electrodes are referenced to contralateral mastoids.

This was corrected.

Pg 8, Line 304: “EEG hemisphere” is unclear; please just say right or left side.

This was corrected.

Pg 16, line 630: Analysis in 5 sec epochs was almost as good as 30 sec, but this does not get much discussion. I think this is an important point and could be expanded upon.

Thank you for this comment. We added 2 sentences to this effect.

Pg 19, line 741: Some clinicians are already confused by the meaning of sleep-onset REM sleep periods (SOREMP), and the author’s use of the term SOREMP will only add to the confusion. I suggest something like “short latency REM sleep”. Also, this should be defined in the main text, not the Supplement.

This was done.

Pg 20, line 770: The authors state that “The use of this novel biomarker will reduce time spent to a standard 8-hour night recording...”, but then this is somewhat contradicted by the last sentence of the paragraph which correctly points out that the MSLT also provides useful information on the propensity for a patient to fall asleep during the day. For now, the MSLT will remain important for diagnosing hypersomnias other than NT1, and I hope this sort of ML PSG analysis does not displace the MSLT in diagnosing NT1.

The term will was replaced by could.

The text still contains a few typos:

Pg 16, line 628: should be “extent”

Pg 19, line 737: should say “...type 1 narcolepsy were...”

Pg 19, line 741: should be “...bears great resemblance...”

Pg 19, line 761: might be clearer as “A PSG with high staging uncertainty...”

Typos have been fixed.

Reviewer #2 (Remarks to the Author):

The authors have addressed all concerns in revised manuscript.

Thank you.

Reviewer #3 (Remarks to the Author):

I want to thank the authors for their responses to my previous comments. Since the exact changes made to the manuscript are not detailed in the rebuttal, I apologize if I have overlooked something that has already been done.

I agree that it will not be necessary to add a multiscorer cohort that includes both sexes (referring also to Table 8), yet appreciate the note of caution added to the discussion.

The added cohorts that replicate the high accuracy on sleep stage and narcolepsy classification further strengthen this work. The authors might consider stressing that this additional replication (also in the healthy controls) further validates their detector also in a non-clinical setting.

This was done. We added a sentence.